# Evaluation and Optimization of the Layout of Community Public Service Facilities for the Elderly: A Case Study of Hangzhou

Yonghua Li [1,2], Qinchuan Ran [1], Song Yao [1,3,*] and Likun Ding [4,*]

1  Department of Regional and Urban Planning, Zhejiang University, Hangzhou 310058, China
2  Zhejiang University Architectural Design and Research Institute Co., Ltd., Hangzhou 310058, China
3  Center for Balance Architecture, Zhejiang University, Hangzhou 310058, China
4  Zhejiang Urban and Rural Planning Design Institute Co., Ltd., Hangzhou 310030, China
*  Correspondence: 22012148@zju.edu.cn (S.Y.); dlk_zju@163.com (L.D.)

**Abstract:** Community public service facilities have a primary supportive role in the health of the elderly. Under the background of global aging, it has become vital to evaluate the elderly-adaptability of their layouts. Based on the supply and demand interaction perspective, this study used the questionnaire-AHP-2SFCA method for this purpose. Firstly, taking the six main districts of Hangzhou as an example, we analyzed the spatial distribution characteristics of the elderly population, and a weight index table of the health importance of public service facilities was constructed using a questionnaire survey and the AHP method. Secondly, the improved 2SFCA was used to analyze the accessibility of public service facilities in Hangzhou, and combined with the weight index table, the elderly-adaptability of public service facilities in the community life circle was comprehensively evaluated. Finally, the demands of the elderly and the supply of public service facilities in the same region were superimposed to study the differential pattern of supply and demand. The results showed the following: (1) The communities with the largest elderly population are mainly concentrated in Shangcheng District, Xiacheng District, the north of Gongshu District, the west of Jianggan District, and the north of Binjiang District. (2) Green space facilities in parks are most important to the health of the elderly, with a weight of 0.46. (3) The overall evaluation results of the community life circle in the study area were good, and the proportion of areas above the medium level was more than 50%. This showed that the concepts of "neighborhood center" and "big community elderly care" in Hangzhou have achieved initial positive results. (4) Based on the interaction between supply and demand, the research area can be divided into four patterns: supply and demand balance, supply shortage, demand gap, and low supply and demand. The results of this study will help to improve the layout and aging-friendly status of community life circle facilities in Hangzhou, and provide information for other aging cities.

**Keywords:** community public service facilities; elderly-adaptability; aging-friendly community; Hangzhou; supply and demand interaction

## 1. Introduction

Population aging is currently a global challenge. The World Health Organization's "Active Aging" Policy Framework states that active aging is the process of optimizing opportunities for health, participation, and security to enhance people's quality of life as they age. The aging global population is the most critical medical and social demographic problem worldwide [1,2]. The number of people aged 60 years or older is expected to reach 2.1 billion by 2050 [3]. Under the background of the increasingly severe global aging problem, the scale of the elderly population is also expanding, and the need for elderly health care is growing [4]. However, the current supply level of public service facilities is yet to be improved accordingly, and there is a massive gap between supply and demand. The mismatch between supply and demand has resulted in problems of health inequity

among the elderly. Addressing this problem is the highest priority for supporting the aging population worldwide.

In 2022, the National Health Commission of the People's Republic of China and other departments jointly issued "the 14th Five-Year Plan for Healthy Aging", proposing that by 2025, health service resources for the elderly should be allocated more rationally, and their health needs should be better met. It is crucial to address the mismatch between supply and demand. Since the public service facilities within the community life circle are more in line with the elderly's trip characteristics of proximity and circle (Figure 1), they have become the primary focus to solve the supply and demand mismatch. The reasonable distribution of community public service facilities has a significant impact on the health of the elderly [5]. A scientific evaluation of the public service facilities for the elderly within the community life circle is an important first step to optimize the allocation of health service resources for the elderly.

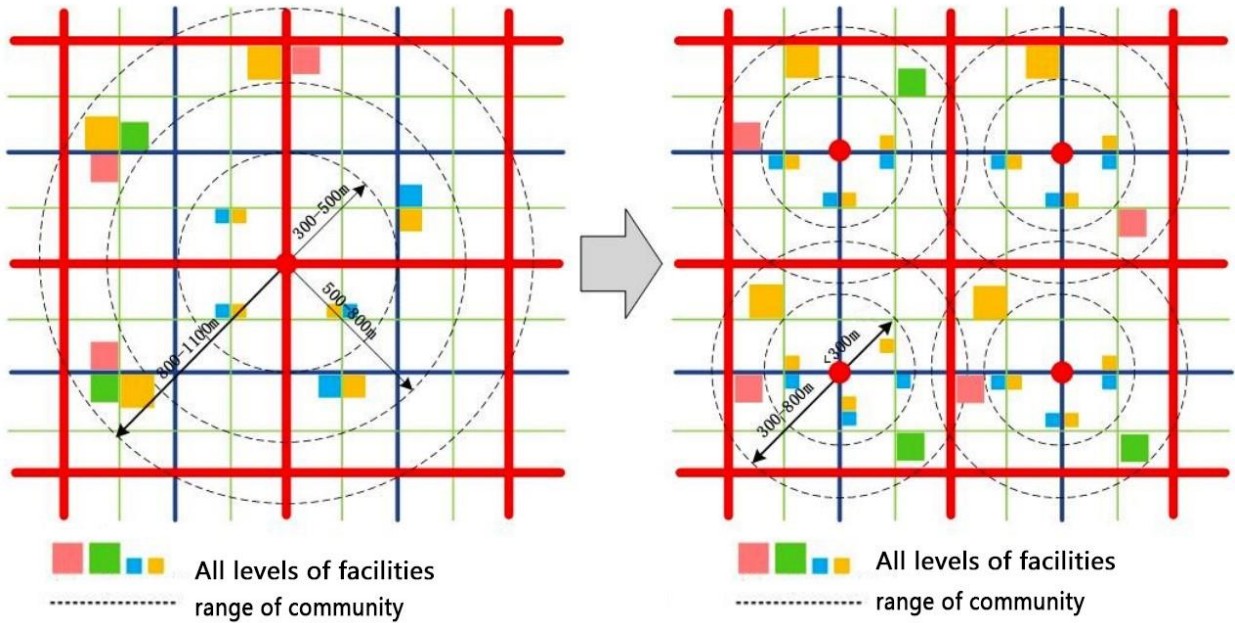

**Figure 1.** Traditional community life circle to an aging-friendly community life circle.

Regarding the resource allocation of health services for the aged, Whitehead put forward the concept of "Health Equity", which linked equity with equal health opportunities for the first time. In order to accurately evaluate the resource allocation of health services for the elderly, an increasing number of researchers have evaluated public service facilities for the aged [6–11]. The research on public service facilities for the elderly has undergone three stages of development [12–24]: (1) Meeting the physiological needs of the elderly (barrier-free facilities and aging-friendly housing in areas with early aging, such as the United States and Japan) [25–40]. (2) Meeting the physical and psychological needs of the elderly (setting up separate facilities for the elderly and creating special public spaces for the elderly) [41–44]. (3) Exploring the diversified and multi-level needs of the elderly [45–48]. The concept of "Health Equity" has been a critical issue in public service facilities for the elderly. The protection of health equity for older people has international consensus [49–52]. Relevant studies have focused on the allocation of health resources, life behaviors, and personal characteristics, and the research can be summarized based on three aspects: age-friendly design, age-friendly communities, and age-friendly cities (Table 1). In the study of public service facilities for the elderly, research has focused on two aspects [53–57]: (1) Based on the physiological and psychological needs of the elderly, exploring the adaptive design of the space environment. (2) Seeking equity in health, services, and benefits on the "scale" of the elderly. Most studies have been conducted from the

perspective of the needs of the elderly population, and few have been conducted from the perspective of the interaction between supply and demand. Because of the contradiction between the growing demand for health care and the supply of existing facilities, it is necessary to evaluate the public service facilities for the elderly in the community from the angle of the interaction between supply and demand.

**Table 1.** Summary of the research focus of "Health Equity".

| Study Classification | Research Focus |
| --- | --- |
| Age-friendly design | Architectural design, public spaces, public facility structures, etc. |
| Age-friendly community | Public service facilities, community internal environment, road traffic planning, settlement conditions, etc. |
| Age-friendly city | Ecological environment, material living environment, spiritual and cultural environment, healthy environment, etc. |

## 2. Study Area and Data

### 2.1. Study Area

This paper considered six districts of the central city of Hangzhou as the object of study, including Xihu District, Gongshu District, Shangcheng District, Xiacheng District, Jianggan District, and Binjiang District (Figure 2). According to the data on Hangzhou's community population in 2019, Hangzhou had 638 communities, with a total area of about 16,850 square kilometers. The number of elderly people aged 60 and above in the six districts of Hangzhou's central city had reached 617,200, and the overall level of aging in the six districts was 22.42%, placing it in the stage of pattern-rate aging. Shangcheng District, Xiacheng District, and Gongshu District had the most significantly aging elderly populations (Figure 3). The central six districts could be classified as having an aging society, with the elderly population showing 'aging, disability, and empty nest' superimposed characteristics.

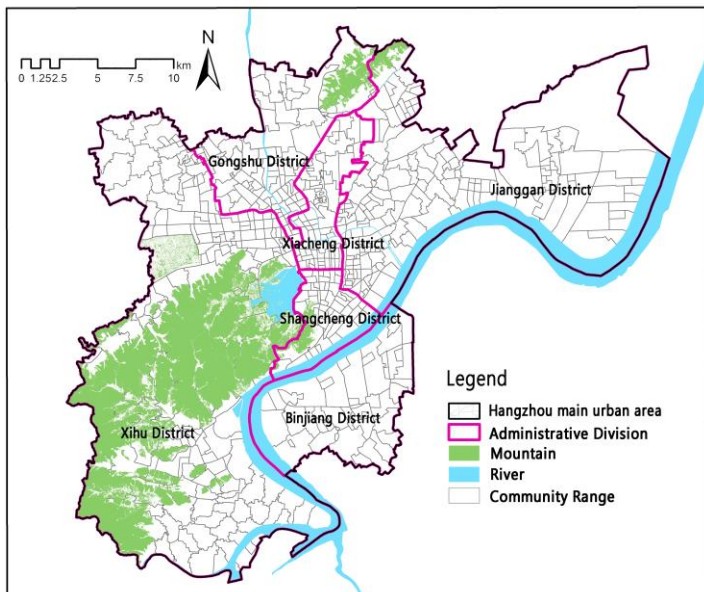

**Figure 2.** Scope of the study area.

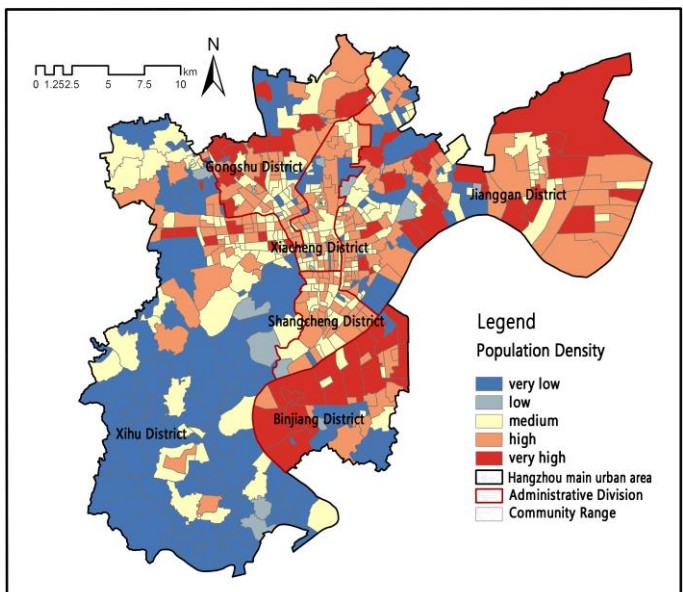

**Figure 3.** Community population distribution in the study area.

*2.2. Data Acquisition*

The community scope of the six districts, the road data of each level, the regional mountain and green space data, the park green space data, and the community population data were provided by the Hangzhou Municipal Platform for Common Geospatial Information Services. The 2019 Hangzhou administrative district scope and primary water system data were provided by the official website of OpenStreetMap at https://www.openstreetmap.org/ accessed on 4 May 2021. Medical and health facilities, elderly care service facilities, sports and fitness facilities, and cultural service facilities data were collected from Baidu Map API. The 2019 community population data provided by the Hangzhou Bureau of Planning and Natural Resources were based on the community as the statistical caliber, including gender, age, education, and other attributes.

*2.3. Data Pre-Processing*

2.3.1. Demand Characteristics of the Elderly

In the yearbook statistics of Hangzhou and the census data of the elderly population, 60 years old is used as the delimitation criterion, so the elderly referred to in this article are 60 years old and over. The degradation of the various functions of the elderly, the decline of social adaptability, and the slow pace of life can lead to loneliness and a sense of inferiority. Maslow's hierarchy of needs theory categorizes human needs into five categories: physiological needs, safety needs, belonging and love, esteem, and self-actualization [49]. The elderly have a lot of leisure time compared to other age groups and are often very willing to interact socially. In the elderly group, the five categories of Maslow's hierarchy of needs can be summarized considering three aspects: social health characteristics, mental health features, and physiological health characteristics (Figure 4) [58].

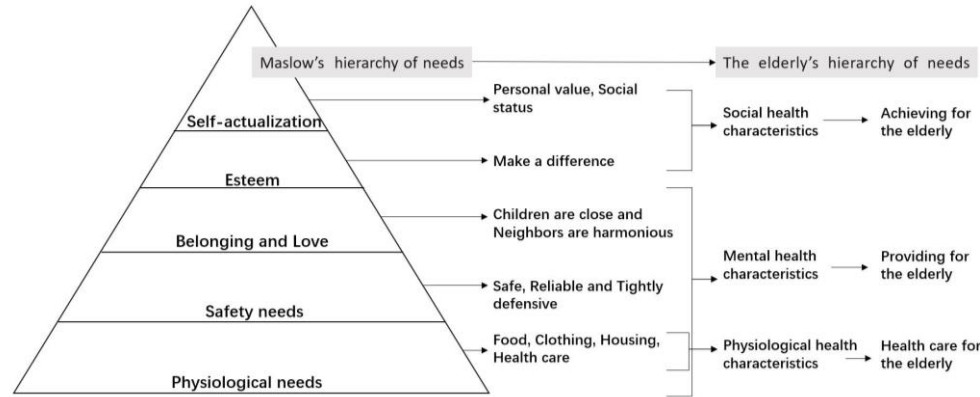

**Figure 4.** Schematic diagram of the health characteristics of the elderly.

### 2.3.2. Layout Characteristics of Community Service Facilities

The concept of the graded construction of life circles was put forward in the "Urban Residential Area Design Standard (2018)" and "Hangzhou Neighborhood Center Planning Research" issued by Hangzhou. The reports point out that most elderly people can tolerate a walking time of 5–10 min, and the available ranges of the 15-min life circle is relatively large. The stratification of a "5-min, 10-min, 15-min life circle" is necessary for the elderly.

Currently, the care service system for the elderly is largely reliant on home-based care services, with community and institutional care services acting as a supplement. Based on the demand characteristics of the elderly and the layout of service facilities, the life circle of the elderly can be divided into four levels: a 5-min basic life circle, a 10-min neighborhood life circle, a 15-min daily life circle, and a more than 15-min extended life circle (Figure 5). With a combination of community facilities at different levels, the service facilities can mainly be divided into five categories: medical and health facilities, elderly public service facilities, sports facilities, cultural facilities, and park facilities. Using POI data, the service facilities were divided into five categories and 14 sub-categories (Table 2).

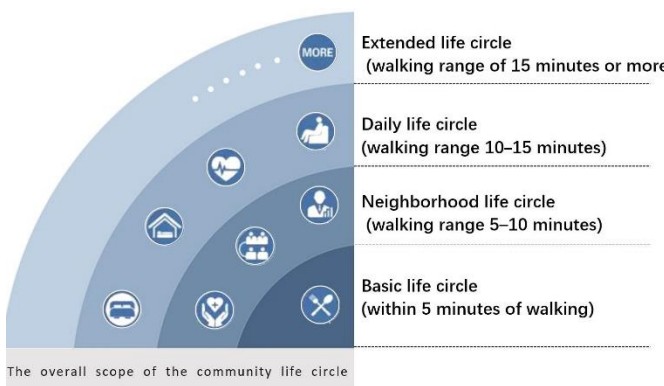

**Figure 5.** Schematic diagram of the hierarchical division of the life circle of the elderly.

**Table 2.** Public service facilities related to the health of the elderly and the corresponding life circle levels.

| The Demands of the Elderly | Category | Subclass | Life Circle Level |
|---|---|---|---|
| Health Care for the Elderly | Medical and Health Facilities | Community health service centers, outpatient departments | 15-min |
| | | Clinics | 10-min |
| | | Community health service stations, pharmacies | 5-min |

Table 2. *Cont.*

| The Demands of the Elderly | Category | Subclass | Life Circle Level |
|---|---|---|---|
| Providing for the Elderly | Elderly Public Service Facilities | Nursing institutions, nursing homes (200–500 beds) | 15-min |
| | | Home care service centers, Starlight Home for the Elderly (5–200 beds) | 10-min |
| | | Community canteen for the elderly | 5-min |
| Achieving for the Elderly | Sports Facilities | Comprehensive gymnasiums, fitness centers | 15-min |
| | | Medium-sized sports pitches (including a variety of pitches) | 10-min |
| | | Outdoor fitness points, chess and card rooms | 5-min |
| | Park Facilities | Large park green spaces (integrated parks) (5–10 ha) | 15-min |
| | | Medium-sized parkland (community parks) (1–5 ha) | 10-min |
| | | Small park green spaces (community amusement parks) (0.4–1 ha) | 5-min |
| | Cultural Facilities | Community cultural activity centers | 15-min |
| | | Community cultural activity stations (library, activity room) | 5-min |

## 3. Methods

### 3.1. Evaluation Framework

A technical flowchart of this study is provided in Figure 6 and consists of three technical steps. Firstly, we analyzed the spatial distribution characteristics of the elderly population, and a weight index table of the health importance of public service facilities was constructed using a questionnaire survey and the AHP method. Secondly, the improved 2SFCA was used to analyze the spatial distribution and accessibility of public service facilities in Hangzhou, and combined with the weight index table, the elderly-adaptability of public service facilities in the community life circle was comprehensively evaluated. Finally, the demands of the elderly and the supply of public service facilities in the same region were superimposed to study the differential pattern of supply and demand.

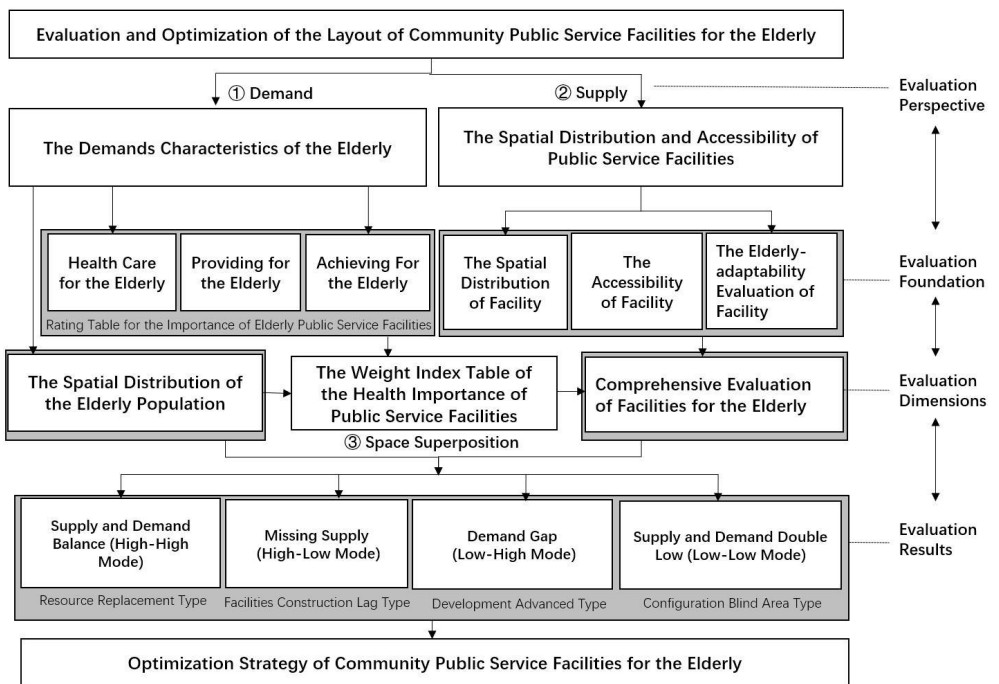

**Figure 6.** Evaluation framework.

*3.2. Questionnaire Method*

The questionnaire method refers to the method of researching in a question-based manner. Researchers use this method of controlled measurement to measure the problem under study and collect reliable data [59–62].

This study collected questionnaire data from the elderly in the main urban area of Hangzhou through a questionnaire survey and field interviews, aiming to obtain information on the actual trip characteristics, facility demand characteristics, and overall satisfaction of the elderly based on their scores and evaluations of the health importance of the public service facilities they used, in order to provide an essential reference for the construction of facility health importance evaluation indicators through an AHP analytic hierarchy.

*3.3. The Two-Step Floating Catchment Area Method*

Rsake first proposed the two-step floating catchment area method (2SFCA). Based on the location of the supply and demand end, as well as their respective scale and demand size, twice the number of searches should be carried out within a certain service distance. Finally, the value of the supply and demand ratio is calculated to obtain the reachability of a point. This method is widely used in the accessibility measurement of medical facilities, parks and green spaces, and various public service facilities [63–67].

The traditional two-step floating catchment area method has some shortcomings. It assumes an equal demand even if one public service facility point is 50 m away from the elderly, and another is 500 m away from the elderly. Although they both may be within the search threshold range of 500 m, in reality, the elderly will choose the closer facility point. The demand degree for a facility will decrease as the distance increases. Therefore, in this study, a Gaussian attenuation function was added to the original supply–demand equation to reflect the attenuation effect of distance on facility supply. Its linear variation trend was consistent with the demand for facility points based on location, which effectively reflected the accessibility difference characteristics within the threshold range, thus enhancing the accuracy of an accessibility evaluation. The calculation formula of the Gaussian function is shown in Equation (1).

$$G_{(d_{ij}d_0)} = \begin{cases} \dfrac{e^{-(\frac{1}{2}) \times (\frac{d}{d_0}ij)^2} - e^{-(\frac{1}{2})}}{1 - e^{-(\frac{1}{2})}}, d_{ij} \leq d_0 \\ 0, d_{ij} > d_0 \end{cases} \tag{1}$$

The centroid of all public service facilities involved in the study was extracted. Due to the large scale of the medium and large park green space, all the entrances and exits were taken as supply points of the green space, all facility points at the supply end were taken as $j$, and the service range of facilities, namely the distance threshold $d_0$, was taken as the search radius to search the collection of demand points within all service radius. $d_{ij}$ was the actual distance between the demand point $i$ and facility supply point $j$. Based on a walking speed of 0.8 m/s for the elderly, the service radius of 5-min, 10-min, and 15-min life circle facilities was 300 m, 500 m, and 800 m, respectively. A classification search was conducted to ensure that the results were basically consistent with actual use.

*3.4. The Analytic Hierarchy Process Method*

The analytic hierarchy process (AHP) sorts and compares all the influencing factors of the research evaluation, builds a reasonable index system, and then determines the mutual importance of each index through expert scoring to form a judgment matrix, before finally calculating the weight of each index according to the matrix [68–72].

In this study, the judgment matrix was divided into two levels. The target level of both levels was "select the most important public service facilities for health", and the scheme level was "different population points in the main urban area of Hangzhou". The index level was divided into two levels, namely medical facilities, elderly care facilities, sports facilities, cultural facilities, and park facilities. The second level evaluated the sub-category

facilities within the category facilities. Experts were asked to score the health importance of facilities for the elderly.

$$\lambda_{\max} = \sum_{i=1}^{n} \frac{[AW]i}{nW_i} \tag{2}$$

Consistency test Formula (2) was used to test the scientific nature of the weight assignment. In the formula, *AW* is the weight value obtained by *i* factor judgment, and $W_i$ is the judgment value of *i* factor. $\lambda_{\max}$ is the maximum eigenvalue of the matrix. The maximum eigenvalues of all matrices were less than 0.1, which met the matrix establishment condition, indicating that the weight assignment was relatively scientific and reasonable.

## 4. Results

We summarized the characteristics of the demand for healthy aging and the layout characteristics of service facilities. Through the questionnaire survey method, we collected and summarized the actual demand characteristics of the elderly, carried out a facility health importance evaluation, and used the analytic hierarchy process (AHP) to construct the facility health importance evaluation index table. From the perspective of supply and demand interaction, the Gaussian two-step floating catchment area (2SFCA) was used to define different service thresholds for different levels of facilities, adding attenuation effects caused by the distance of use, and we also measured the service level of different types of facilities. On this basis, the weight index of health importance was added, and the comprehensive aging-friendly evaluation results of the public service facilities in the community life circle were obtained.

### 4.1. Analysis of Demand for Community Public Service Facilities among the Elderly

4.1.1. Structure and Spatial Distribution of the Elderly Population

According to the age structure attributes of each community, the number of people aged 60–69, 70–79, and over 80 was added to obtain a distribution map of the elderly population in each community. From the perspective of layout, the communities with the largest elderly population were concentrated in the upper urban area, the whole lower urban area, the northern part of Gongshu District, the western part of Jianggan District, and the northern part of Binjiang District (Figure 7).

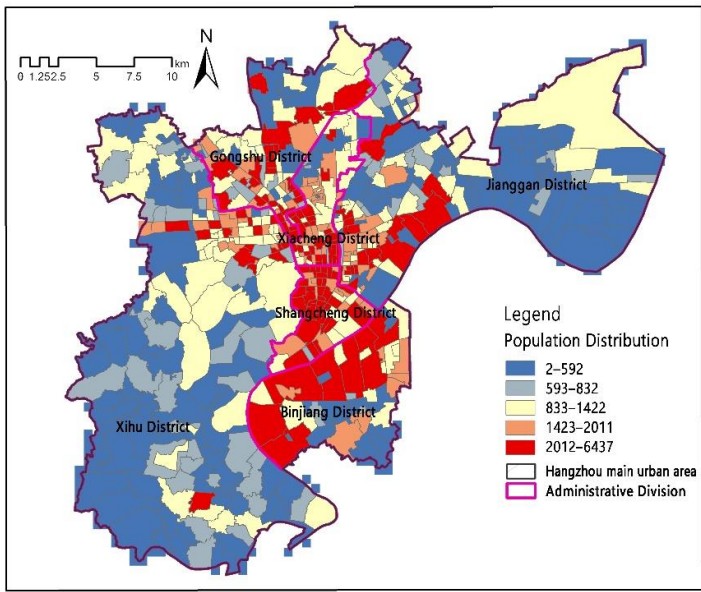

**Figure 7.** The distribution of the elderly population.

The scale and hierarchy of the community life circle in this paper were detailed, with 2073 population grids (population cells) of 600 m × 600 m. After grid segmentation, the

original communities of different sizes formed a uniform population demand centroid in the cell, which can better meet the needs of subsequent fine-matching research. After grid segmentation, the number of elderly people in each cell decreased, and the overall distribution of the elderly population showed a decreasing trend from the center to the edge (Figure 8).

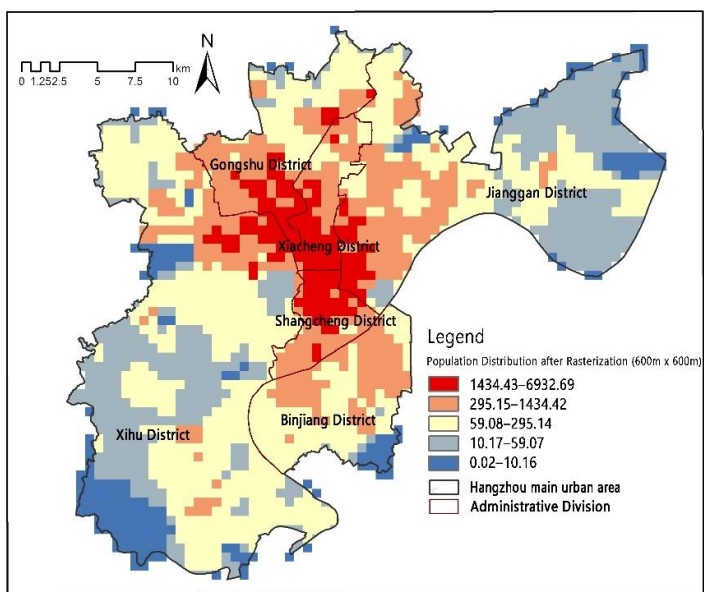

**Figure 8.** The distribution of the elderly population after community rasterization (600 m × 600 m).

### 4.1.2. Weight Coefficient Table of Importance of Health Public Service Facilities

Through the above verification steps of the AHP analytic hierarchy method, the final public service facility health importance weight index table was obtained (Table 3). Park facilities were given the most critical health weight, followed by medical and health facilities, elderly care service facilities, sports facilities, and cultural facilities. This was consistent with the content of the questionnaire survey results and interviews with the elderly, and the hierarchical importance evaluation played a vital role in the evaluation of the suitability of community life circles and elderly care demand facilities. Through calculation, the maximum feature values of all matrices in this study were found to be less than 0.1, meeting the conditions for matrix establishment and indicating that the weight distribution was relatively scientific and reasonable.

**Table 3.** Weight coefficient table of the importance of health public service facilities.

| Category | Weight Coefficient | Subcategories | Importance | Weight Coefficient | Normalized Weight |
|---|---|---|---|---|---|
| Medical and Health Facilities | 0.23 | Community health service centers, outpatient departments | 3.90 | 0.58 | 0.13 |
| | | Clinics | 2.40 | 0.11 | 0.03 |
| | | Community health service stations, pharmacies | 3.50 | 0.31 | 0.07 |
| Elderly Public Service Facilities | 0.15 | Nursing institutions, nursing homes (200–500 beds) | 2.30 | 0.10 | 0.01 |
| | | Home care service centers, Starlight Home for the Elderly (5–200 beds) | 3.30 | 0.60 | 0.09 |
| | | Community canteens for the elderly | 2.50 | 0.30 | 0.04 |
| Sports Facilities | 0.10 | Comprehensive gymnasiums, fitness centers | 2.10 | 0.08 | 0.01 |
| | | Medium-sized sports pitches (including a variety of pitches) | 2.20 | 0.15 | 0.02 |
| | | Outdoor fitness points, chess and card rooms | 3.00 | 0.77 | 0.08 |

**Table 3.** *Cont.*

| Category | Weight Coefficient | Subcategories | Importance | Weight Coefficient | Normalized Weight |
|---|---|---|---|---|---|
| Park Facilities | 0.46 | Large park green spaces (integrated parks) (5–10 ha) | 3.00 | 0.16 | 0.08 |
| | | Medium-sized parkland (community parks) (1–5 ha) | 3.80 | 0.54 | 0.25 |
| | | Small park green space (community amusement parks) (0.4–1 ha) | 3.40 | 0.30 | 0.14 |
| Cultural Facilities | 0.05 | Community cultural activity centers | 2.80 | 0.80 | 0.04 |
| | | Community cultural activity stations (library, activity room) | 2.00 | 0.20 | 0.01 |

*4.2. Evaluation of Community Public Service Facilities*

4.2.1. Spatial Distribution Characteristics of Public Service Facilities

The spatial distribution of public service facilities in the six districts of the central city of Hangzhou showed a dense and gradually decreasing density in the central old urban areas, such as the upper urban area, the lower urban area, and the Gongshu District (Figures 9–13). In addition to the linear distribution of park green space facilities along the banks of the Qiantang River and the Beijing-Hangzhou Grand Canal, the rest were similar to the distribution of urban population density. However, there were differences in the distribution of different types of facilities in the 5-min, 10-min, and 15-min life circles.

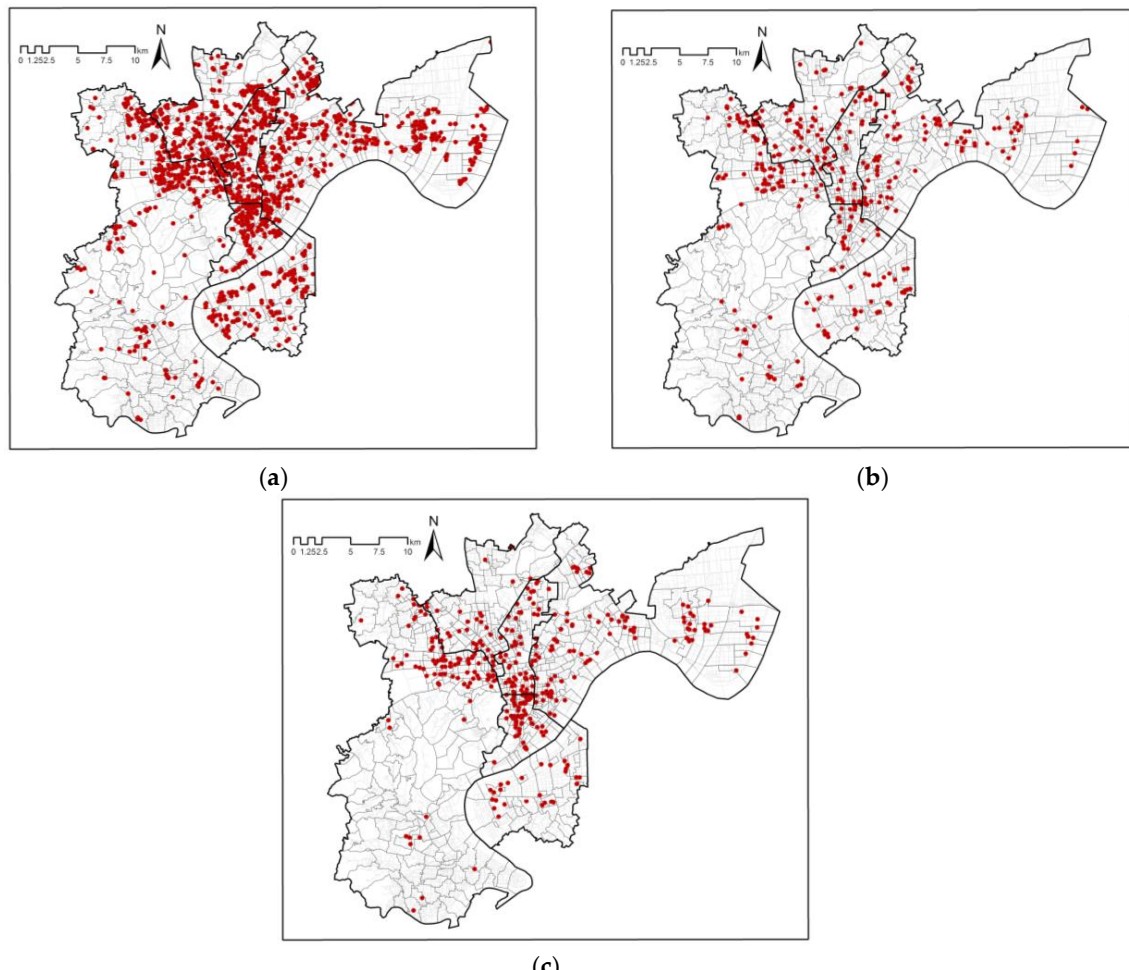

(a)

(b)

(c)

**Figure 9.** Distribution of medical and health facilities in the study area. (**a**) 5-min life circle level, (**b**) 10-min life circle level, (**c**) 15-min life circle level.

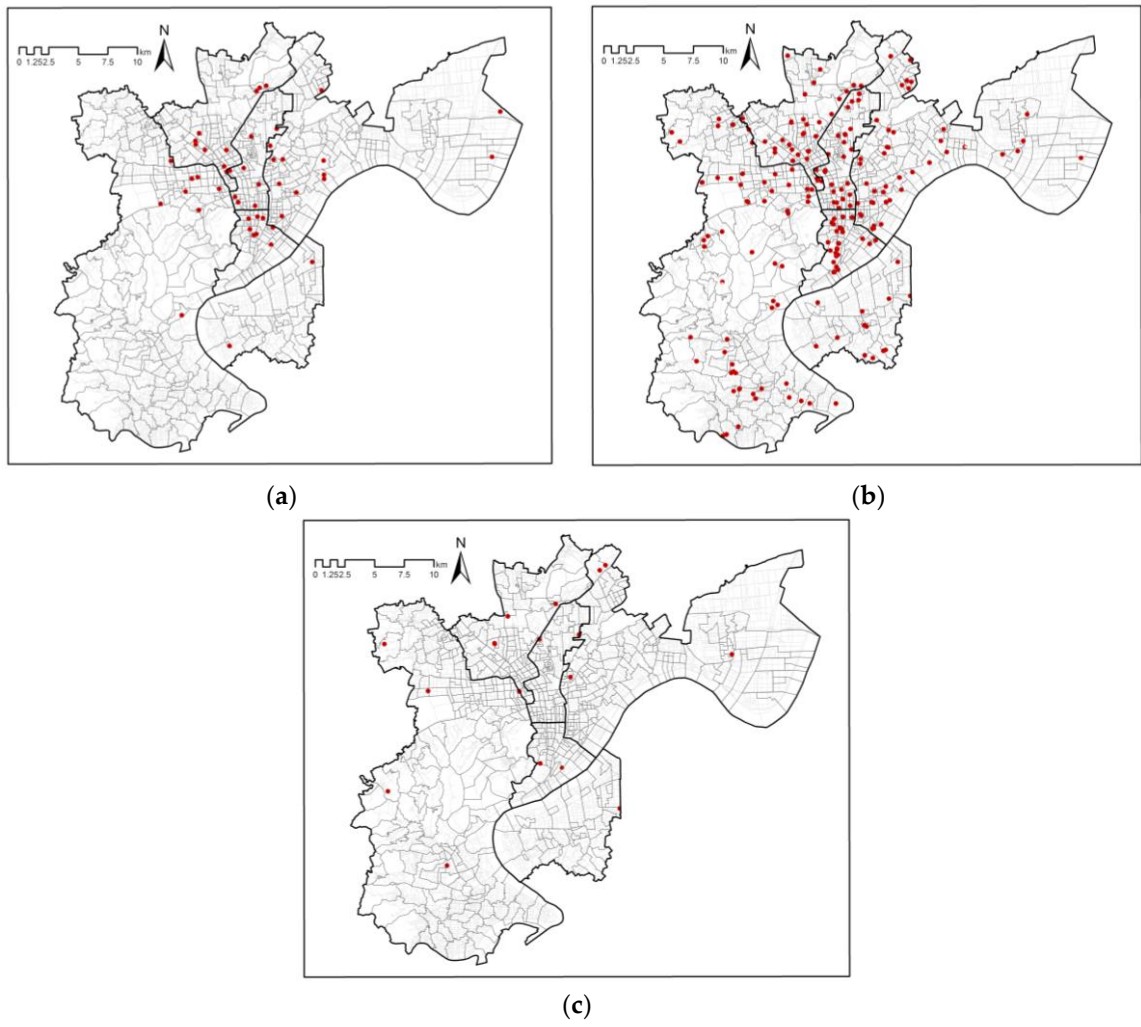

**Figure 10.** The distribution map of pension service facilities in the study area. (**a**) 5-min life circle level, (**b**) 10-min life circle level, (**c**) 15-min life circle level.

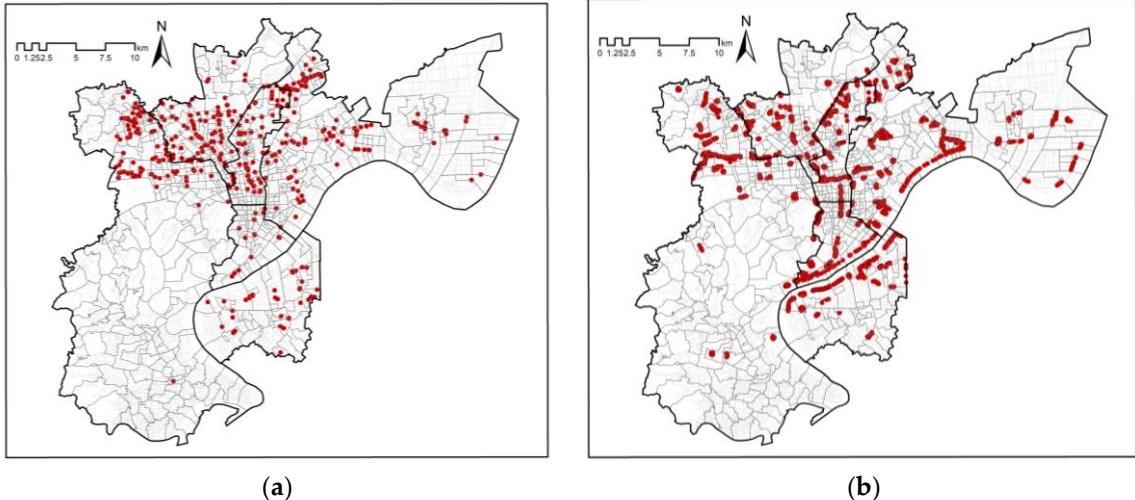

**Figure 11.** *Cont.*

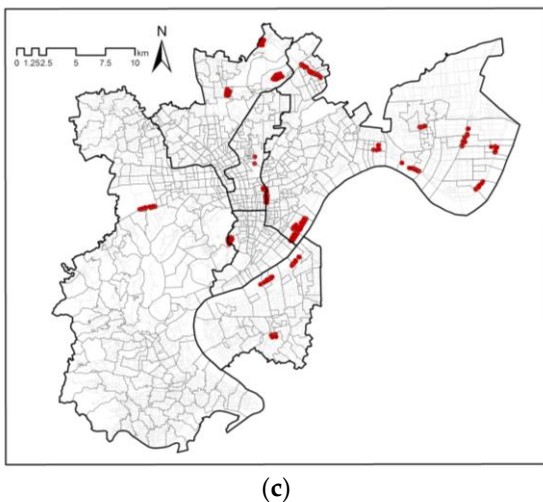

(**c**)

**Figure 11.** Distribution map of park green space facilities in the study area. (**a**) 5-min life circle level, (**b**) 10-min life circle level, (**c**) 15-min life circle level.

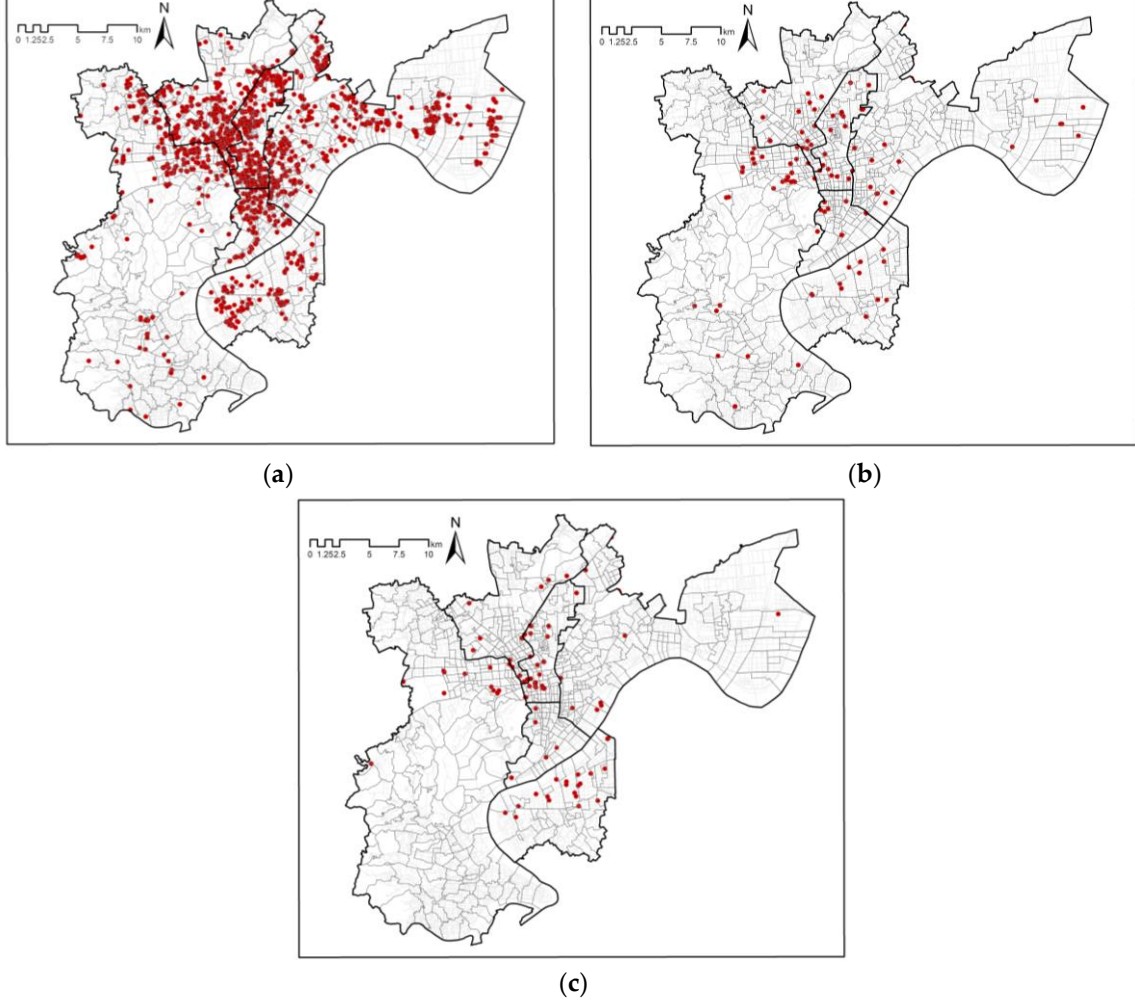

**Figure 12.** Distribution map of sports facilities in the study area. (**a**) 5-min life circle level, (**b**) 10-min life circle level, (**c**) 15-min life circle level.

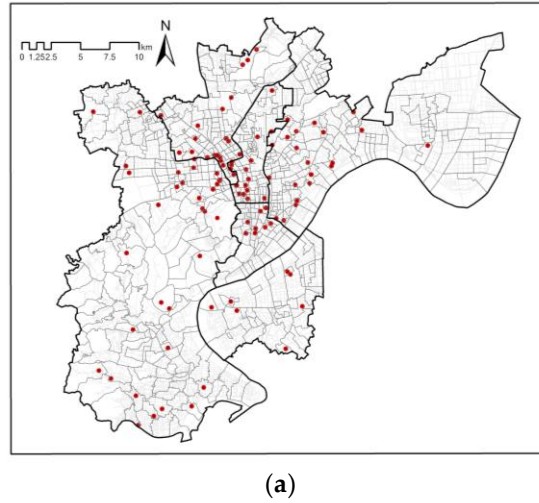 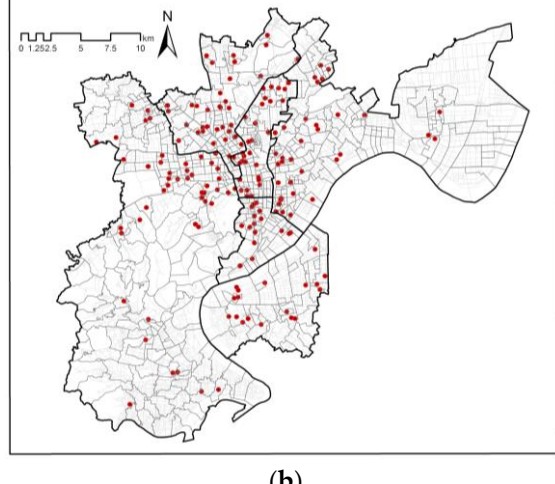

(**a**)          (**b**)

**Figure 13.** Distribution of cultural service facilities in the study area. (**a**) 5-min life circle level, (**b**) 15-min life circle level.

#### 4.2.2. Facilities Accessibility Classification Evaluation

In terms of "Medical and Health Facilities" (Table 4), from the perspective of the spatial distribution of accessibility, the overall accessibility of the 5-min life circle showed the characteristics of dispersion and fragmentation, and the overall accessibility of the 10-minute life circle showed the characteristics of zonal continuity in the north of the main urban area. The 15-min life circle showed almost total coverage of good accessibility in the main urban centers of Shangcheng District, Xiacheng District, the eastern part of Xihu District, and the western part of Jianggan District. The distribution density of community health centers and outpatient departments in these areas was much higher than that in the peripheral areas of the city (Figure 14).

For "Elderly Public Service Facilities" (Table 5), the spatial distribution and quantity showed significant inequity, with the 5-min and 15-min life circle levels showing a sporadic dotted distribution due to the small number of facilities and the failure to reach uniform accessibility coverage. The 10-min life circle level had better accessibility in the central city (Figure 15).

For "Park Facilities" (Table 6), the spatial distribution of accessibility was good overall, with the 15-min life circle level showing an obvious point-like layout and the 10-min life circle level showing a good accessibility distribution in Jianggan District, Binjiang District, Gongshu District, Shangcheng District, and Haicheng District, along with a relatively even spatial distribution (Figure 16). However, the density of small parks and green areas still needs to be increased.

**Table 4.** Accessibility statistics of medical and health facilities by level of life circle.

| Medical and Health Facilities | Very High Reachability Cells (pcs) | Percentage (%) | Higher Accessibility Cells (pcs) | Percentage (%) | Highly Accessible Cells (pcs) |
|---|---|---|---|---|---|
| 5-min life circle (community health service station, pharmacy) | 27 | 1.3 | 263 | 12.7 | 14 |
| 10-min life circle (clinic) | 54 | 2.6 | 86 | 4.1 | 6.7 |
| 15-min life circle (community health service center, outpatient department) | 19 | 0.9 | 136 | 6.6 | 7.5 |
| Integrated community life circle | 34 | 1.6 | 301 | 14.5 | 16.1 |

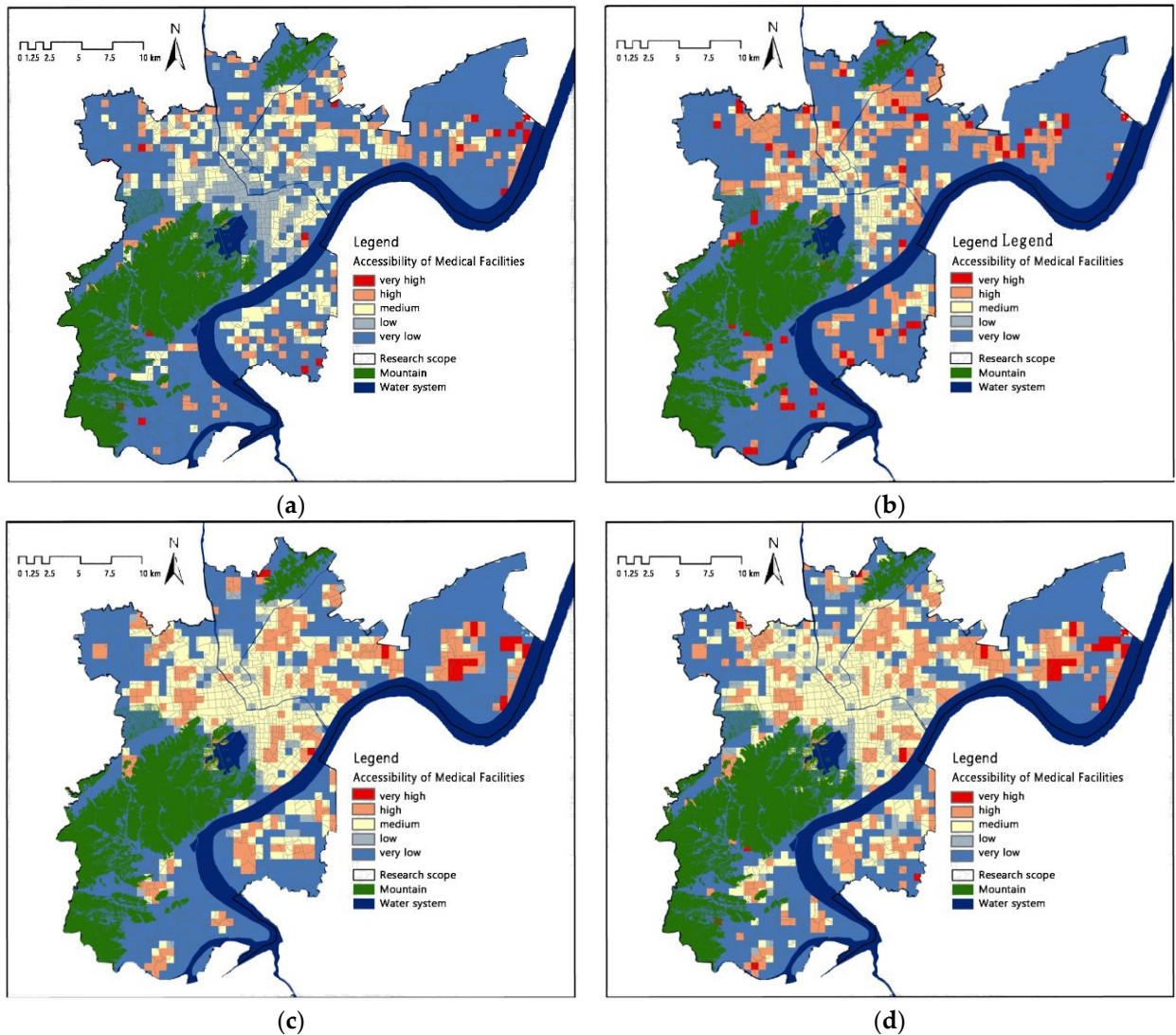

**Figure 14.** Spatial distribution map of the accessibility evaluation of medical and health facilities at each life circle level. (**a**) 5-min life circle level, (**b**) 10-min life circle level, (**c**) 15-min life circle level, (**d**) Comprehensive community life circle level.

**Table 5.** Accessibility statistics for elderly public service facilities at each life circle level.

| Elderly Public Service Facilities | Very High Reachability Cells (pcs) | Percentage (%) | Higher Accessibility Cells (pcs) | Percentage (%) | Highly Accessible Cells (pcs) |
|---|---|---|---|---|---|
| 5-min life circle (community senior cafeteria) | 7 | 0.3 | 6 | 0.3 | 0.6 |
| 10-min life circle (5–200 bed home care service center) | 6 | 0.3 | 108 | 5.2 | 5.5 |
| 15-min life circle (200–500 bed elderly care institution) | 13 | 0.6 | 44 | 2.1 | 2.7 |
| Integrated community life circle | 6 | 0.3 | 110 | 5.3 | 5.6 |

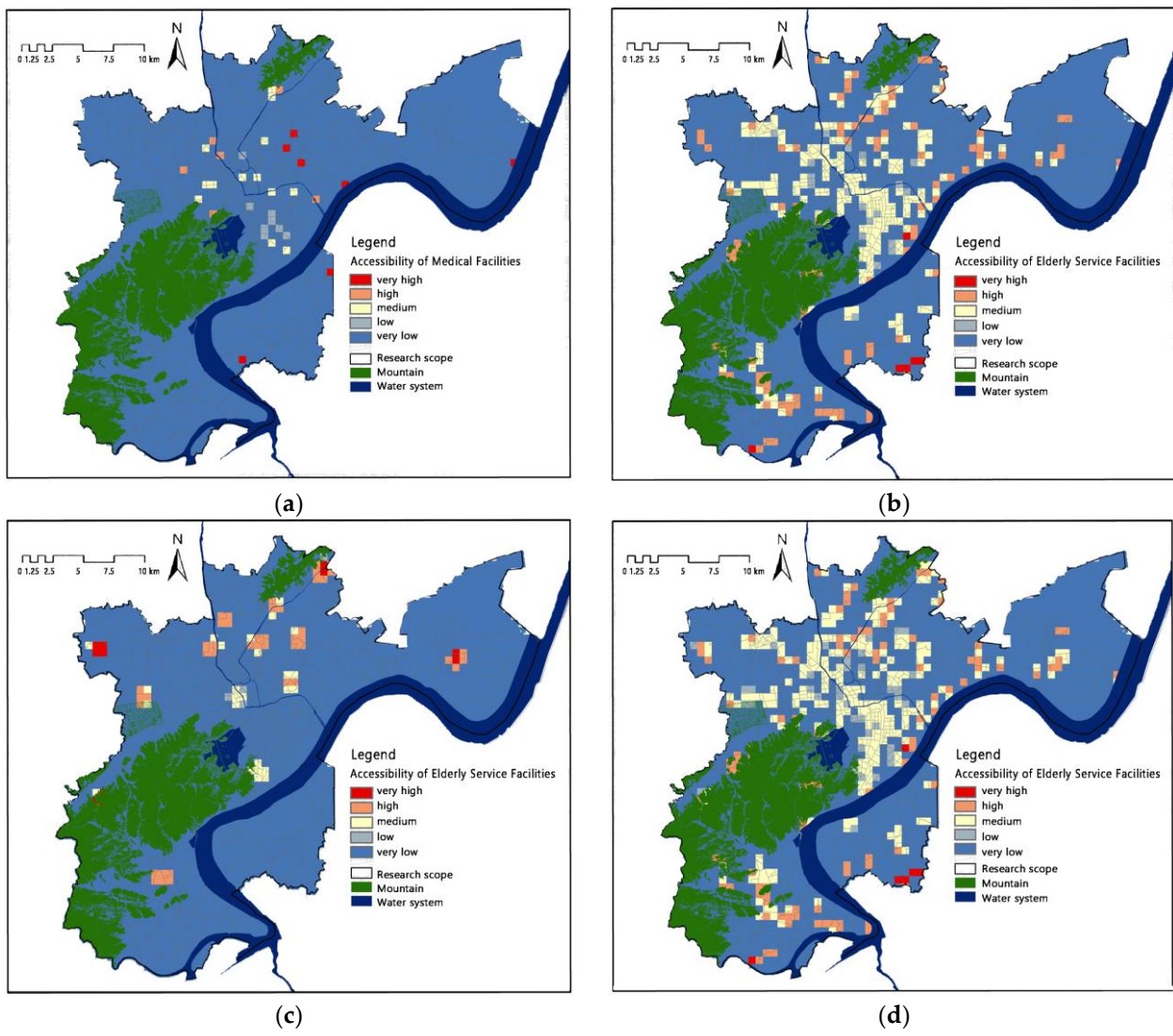

**Figure 15.** Spatial distribution map of the accessibility evaluation of elderly public service facilities at each life circle level. (**a**) 5-min life circle level, (**b**) 10-min life circle level, (**c**) 15-min life circle level, (**d**) Comprehensive community life circle level.

**Table 6.** Accessibility statistics of park facilities at each life circle level.

| Park Facilities | Very High Reachability Cells (pcs) | Percentage (%) | Higher Accessibility Cells (pcs) | Percentage (%) | Highly Accessible Cells (pcs) |
|---|---|---|---|---|---|
| 5-min life circle (small green space) | 10 | 0.5 | 75 | 3.6 | 4.1 |
| 10-min life circle (medium-sized green space) | 25 | 1.2 | 275 | 13.3 | 14.5 |
| 15-min life circle (large green space) | 27 | 1.2 | 107 | 5.1 | 6.4 |
| Integrated community life circle | 26 | 1.3 | 282 | 13.6 | 14.9 |

For "Sports Facilities" (Table 7), there were significant issues in the spatial layout. Except for the 5-min life circle, which covers a wide area, the rest of the circle was concentrated in the central part of the main city, and there was no trend of external radiation (Figure 17).

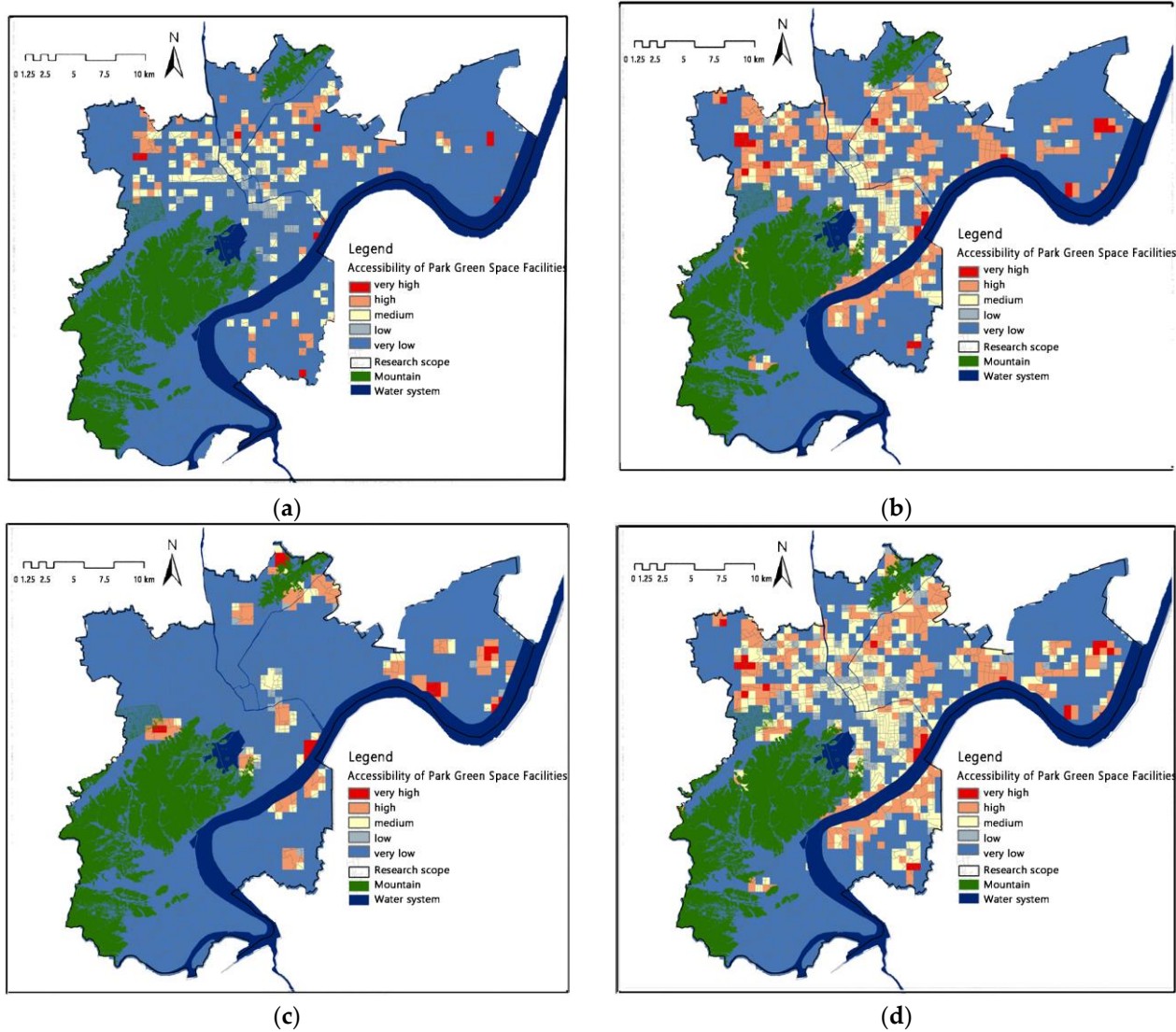

**Figure 16.** Spatial distribution map of the accessibility evaluation of park facilities at each life circle level. (**a**) 5-min life circle level, (**b**) 10-min life circle level, (**c**) 15-min life circle level, (**d**) Comprehensive community life circle level.

**Table 7.** Accessibility statistics of sports facilities by level of life circle.

| Sports Facilities | Very High Reachability Cells (pcs) | Percentage (%) | Higher Accessibility Cells (pcs) | Percentage (%) | Highly Accessible Cells (pcs) |
|---|---|---|---|---|---|
| 5-min life circle (outdoor fitness spots, small activity venues) | 6 | 0.3 | 74 | 3.6 | 3.9 |
| 10-min life circle (medium-sized sports activity sites) | 17 | 0.8 | 86 | 4.1 | 4.9 |
| 15-min life circle (comprehensive gymnasium, national fitness center) | 10 | 0.5 | 123 | 5.9 | 6.4 |
| Integrated community life circle | 14 | 0.7 | 269 | 13 | 13.7 |

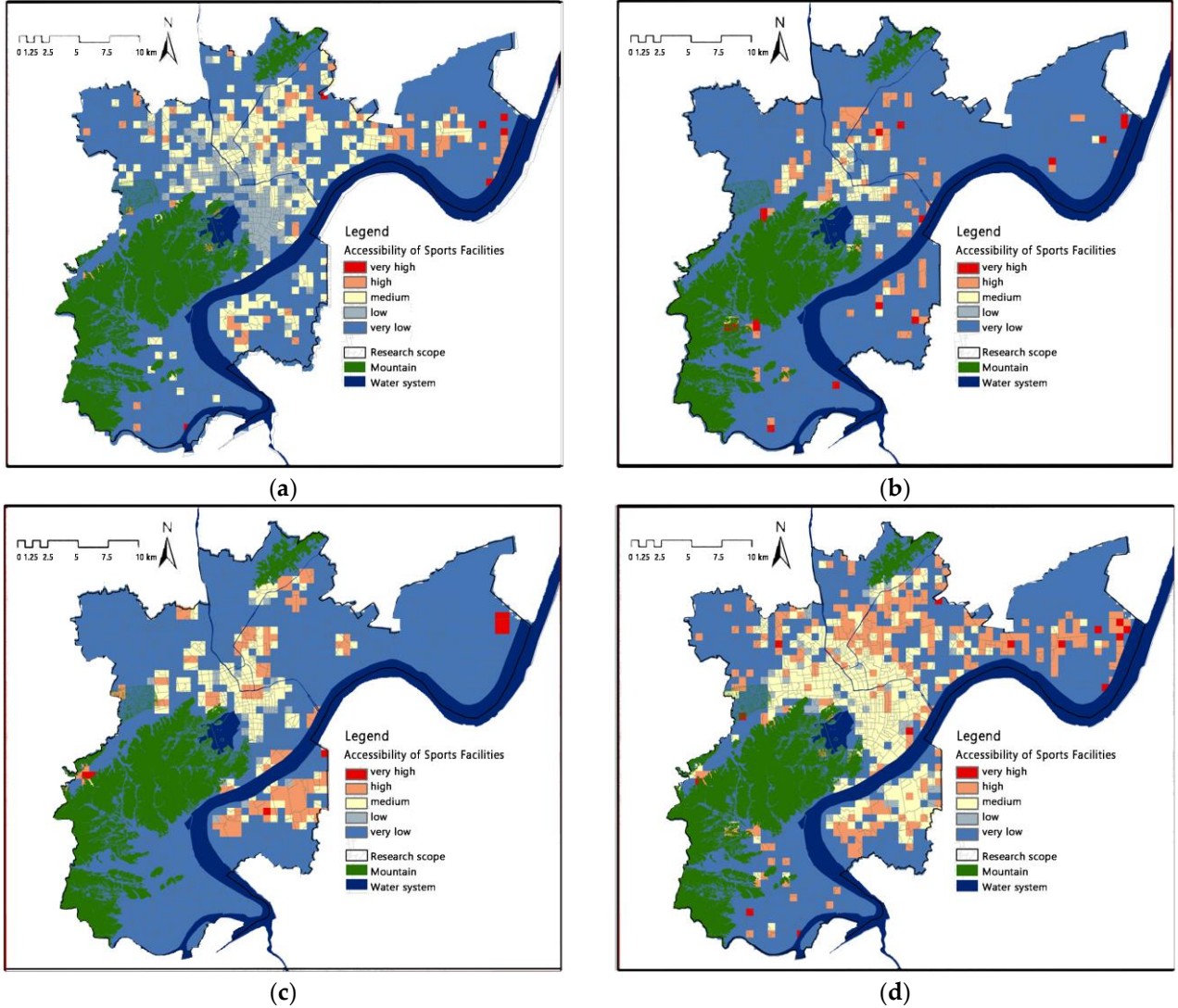

**Figure 17.** Spatial distribution map of the accessibility evaluation of sports facilities at each life circle level. (**a**) 5-min life circle level, (**b**) 10-min life circle level, (**c**) 15-min life circle level, (**d**) Comprehensive community life circle level.

For "Cultural Facilities" (Table 8), the overall accessibility of cultural service facilities in the 5-min life circle was low and showed a scattered distribution. In contrast, the 15-min life circle level had high accessibility and a central gathering and wedge-shaped outward distribution characteristic (Figure 18).

**Table 8.** Accessibility statistics of cultural facilities by life circle level.

| Cultural Facilities | Very High Reachability Cells (pcs) | Percentage (%) | Higher Accessibility Cells (pcs) | Percentage (%) | Highly Accessible Cells (pcs) |
|---|---|---|---|---|---|
| 5-min life circle (community cultural activity station (library, activity room)) | 6 | 0.3 | 23 | 1.1 | 1.4 |
| 15-min life circle (community cultural activity center) | 20 | 1 | 288 | 13.9 | 14.9 |
| Integrated community life circle | 24 | 1.2 | 302 | 14.6 | 15.8 |

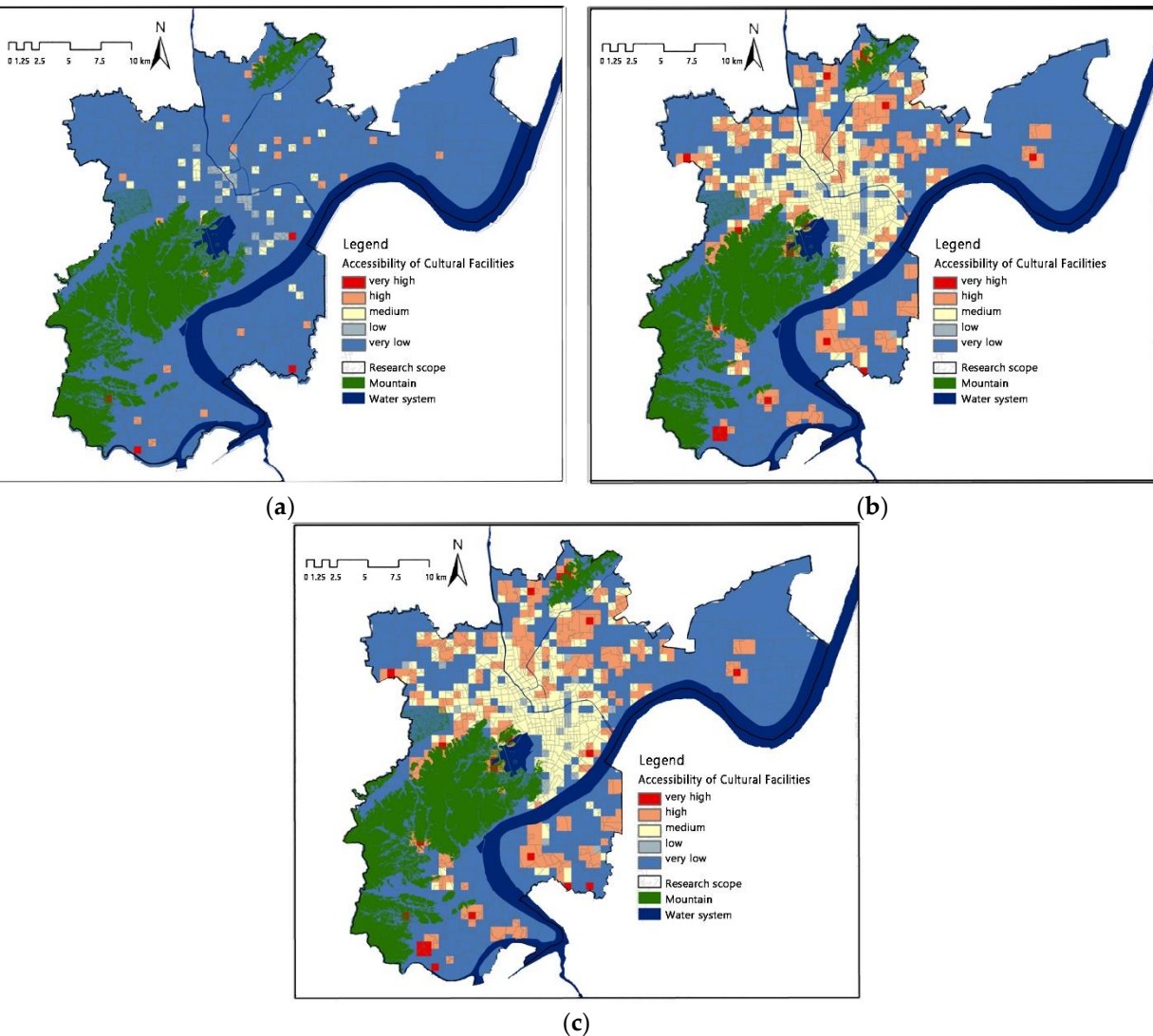

**Figure 18.** Spatial distribution map of accessibility evaluation of cultural facilities at each life circle level. (**a**) 5-min life circle level, (**b**) 15-min life circle level, (**c**) Comprehensive community life circle level.

*4.3. Comprehensive Evaluation of Community Service Facilities under the Interaction of Supply and Demand*

4.3.1. "Health Care for the Elderly, Providing for the Elderly, Achieving for the Elderly" Comprehensive Evaluation of Facilities

Using the health importance evaluation indicators of the elderly for specific life circle level facilities, a comprehensive evaluation of the elderly-oriented layout of the three major types of public service facilities, "Health Care for the Elderly, Providing for the Elderly, Achieving for the Elderly" was conducted to provide more targeted layout suggestions for diversified health care for the elderly. "Health Care for the Elderly" refers to guaranteeing the physical health of the elderly. From the statistical results, the layout of medical facilities in the main urban area of Hangzhou is good (Table 9). From the perspective of the spatial distribution, the areas with low suitability for the elderly are mainly concentrated near the edge of the main urban area, with a low population density, insufficient basic medical facilities, and a low density of community health service stations or community health centers. The evaluation results of layout suitability for the elderly in this area were not ideal (Figure 19). The elderly population density is higher in the areas with medium and lower suitability for the elderly, and the demand for medical facilities is also relatively

large. The elderly population in areas with a more favorable evaluation of layout suitability for the elderly is also relatively large. In these areas, due to the plentiful specialist clinics and community health service stations, the evaluation results of layout suitability for the elderly were better. "Providing for the Elderly" mainly refers to pension service facilities. The layout of pension service facilities in the main urban area of Hangzhou is not ideal for the elderly (Table 10). From the perspective of the spatial distribution, improvement is needed. The areas with a good evaluation of old-age service facilities showed a trend of distribution along the line and are contiguous in the center of the main urban area, but only some sporadic communities in the marginal areas of the central city are within the service scope of old-age facilities (Figure 19). "Achieving for the Elderly" refers to meeting the spiritual and cultural needs of most elderly people (Table 11). From the perspective of spatial distribution, the comprehensive evaluation of elderly-oriented facilities in the main urban area of Hangzhou was high in the center and low in the periphery (Figure 19). In summary, a complete grading system for the pension system still needs to be developed. The urban park green space system should increase the number and density of community parks. At the same time, the configuration system of comprehensive and community parks should be adjusted to change the original single distribution characteristics along the river.

**Table 9.** "Health Care for the Elderly" evaluation of the suitability of the layout of facilities.

| "Health Care for the Elderly" Evaluation of the Suitability of the Layout of Facilities | Very High Values | High Values | Medium Value | Low Value | Very Low Value |
| --- | --- | --- | --- | --- | --- |
| Population cell (pcs) | 34 | 301 | 417 | 112 | 1209 |
| Percentage of population cells (%) | 1.6 | 14.5 | 20.1 | 5.4 | 58.3 |

**Table 10.** "Providing for the Elderly" evaluation of the suitability of the layout of facilities.

| "Providing for the Elderly" Evaluation of the Suitability of the Layout of Facilities | Very High Values | High Values | Medium Value | Low Value | Very Low Value |
| --- | --- | --- | --- | --- | --- |
| Population cell (pcs) | 6 | 110 | 226 | 37 | 1694 |
| Percentage of population cells (%) | 0.3 | 5.3 | 10.6 | 1.8 | 81.7 |

**Table 11.** Evaluation results of the layout of facilities for the elderly.

| "Achieving for the Elderly" Evaluation of the Suitability of the Layout of Facilities | Very High Values | High Values | Medium Value | Low Value | Very Low Value |
| --- | --- | --- | --- | --- | --- |
| Population cell (pcs) | 31 | 362 | 409 | 232 | 1039 |
| Percentage of population cells (%) | 1.5 | 17.5 | 19.8 | 11.2 | 50.1 |

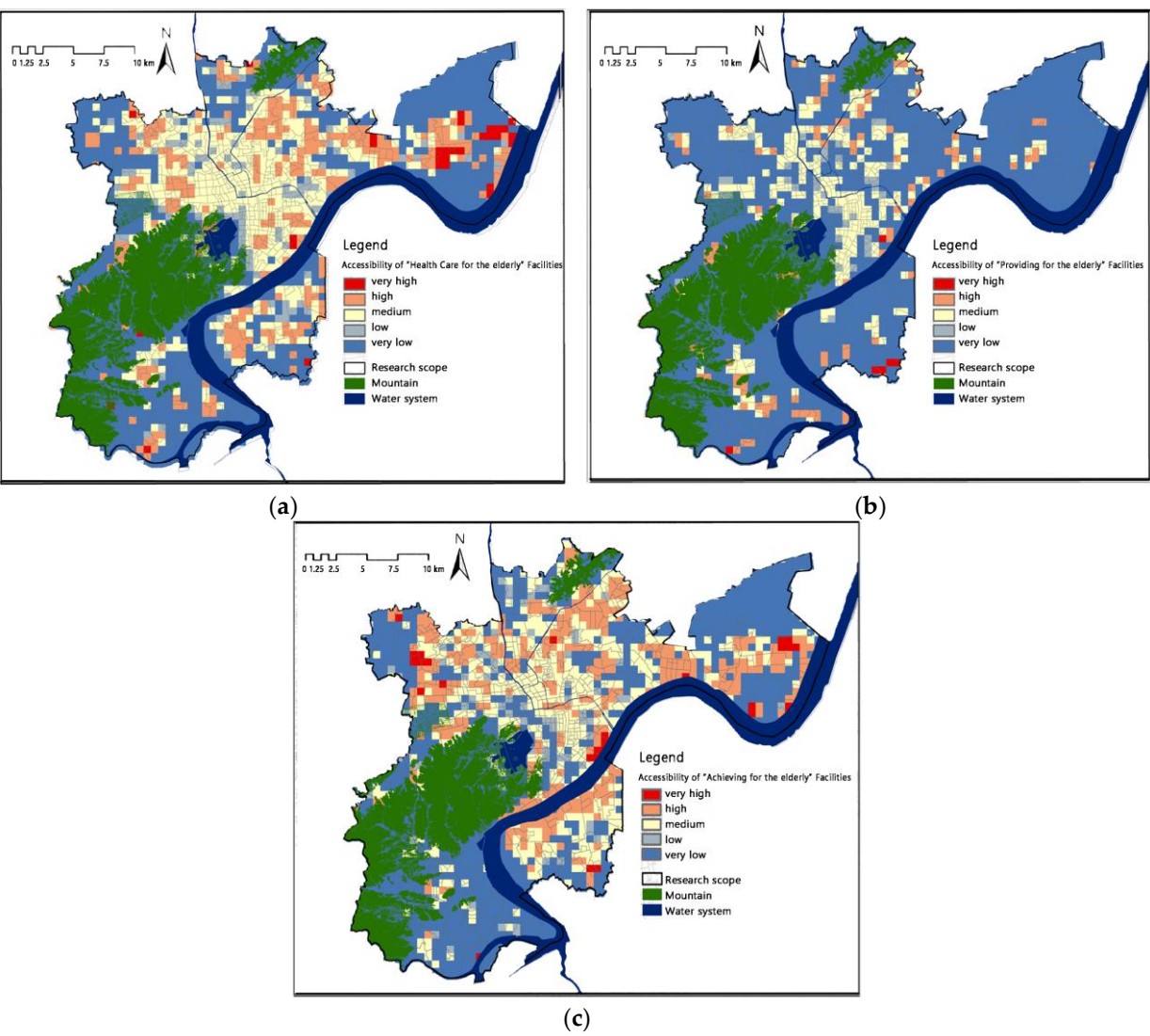

**Figure 19.** The spatial distribution of facility layout comprehensive evaluation. (**a**) "Health Care for the Elderly", (**b**) "Providing for the Elderly", (**c**) "Achieving for the Elderly".

4.3.2. Community Life Circle of Public Service Facilities: Elderly-Adaptability Evaluation

At the 5-min life circle level (Table 12), the cells with high suitability for the elderly and above were found to be distributed in communities with relatively new constructions and ample facilities and resources. The elderly people in these areas are able to reach the required facilities relatively easily by walking. The elderly population with extremely high density in the middle-equivalent cell distribution requires more investment in facilities to meet their actual needs. The low aging-friendly areas are mainly concentrated on the edge of the urban area, with a low population density and relatively backward public service facilities (Figure 20).

**Table 12.** 5-min life circle elderly-adaptability evaluation results.

| 5-Minute Life Circle Elderly-Adaptability Evaluation | Very High Values | High Values | Medium Value | Low Value | Very Low Value |
|---|---|---|---|---|---|
| Population cell (pcs) | 18 | 165 | 370 | 129 | 1391 |
| Percentage of population cells (%) | 0.9 | 8 | 17.8 | 6.2 | 67.1 |

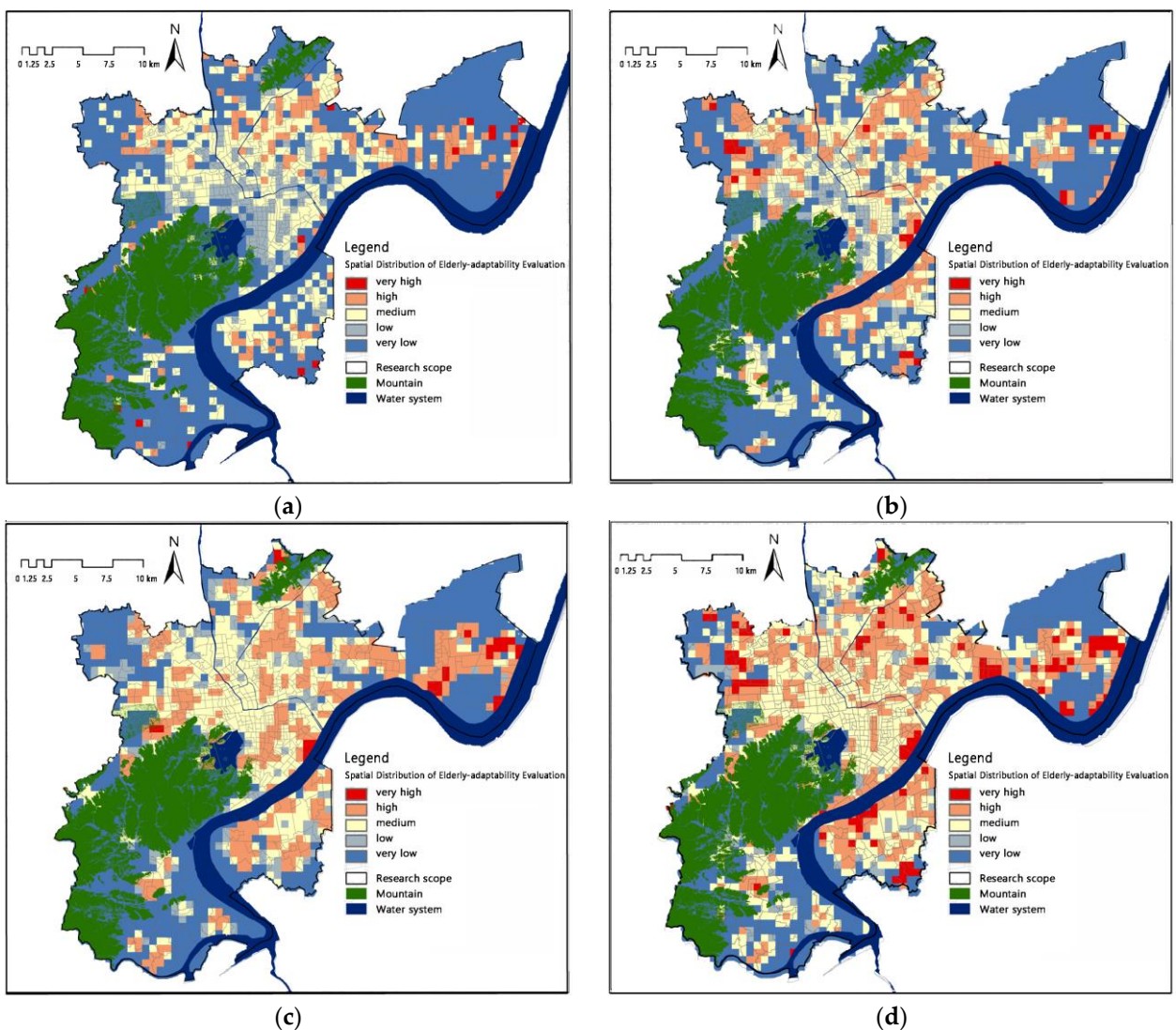

**Figure 20.** Spatial distribution of the elderly-adaptability evaluation. (**a**) 5-min life circle, (**b**) 10-min life circle, (**c**) 15-min life circle, (**d**) Comprehensive evaluation of community life circle.

At the 10-min life circle level (Table 13), the 10-min life circle in the main urban area of Hangzhou is generally better for the elderly. There are ample park green space systems, clinics, and various stadium venues around the cells. These facilities can meet the needs of the elderly in terms of activities, fitness, and medical consultation. Most areas with a low suitability for the elderly are still at the edge of the urban area, and the supporting facilities are seriously lacking (Figure 20).

**Table 13.** 10-min life circle elderly-adaptability evaluation results.

| 10-Minute Life Circle Elderly-Adaptability Evaluation | Very High Values | High Values | Medium Value | Low Value | Very Low Value |
|---|---|---|---|---|---|
| Population cell (pcs) | 26 | 302 | 396 | 127 | 1222 |
| Percentage of population cells (%) | 1.3 | 14.6 | 19.1 | 6.1 | 58.9 |

At the 15-min life circle level (Table 14), the overall evaluation in the main urban area of Hangzhou was relatively good, and there were almost no areas with poor evaluation

results in the old urban areas, such as the lower urban area, the upper urban area, and the Gongshu District. The overall evaluation results of suitability for the elderly were medium and above, although shortcomings remained in some areas on the edge of the city. A large difference in the level of suitability for the elderly was noted (Figure 20).

**Table 14.** 15-min life circle elderly-adaptability evaluation results.

| 15-Minute Life Circle Elderly-Adaptability Evaluation | Very High Values | High Values | Medium Value | Low Value | Very Low Value |
|---|---|---|---|---|---|
| Population cell (pcs) | 36 | 405 | 418 | 147 | 1067 |
| Percentage of population cells (%) | 1.8 | 19.5 | 20.7 | 7.1 | 51.5 |

The evaluation results for public service facilities suitable for the elderly in the 5-min, 10-min, and 15-min life circles were obtained by superposition. The overall evaluation results of the community life circle in the main urban area of Hangzhou were relatively good, and the proportion of areas above the middle equivalent was more than 50% (Table 15). This showed that the implementation of the concepts of "neighborhood center" and "big community pension" in Hangzhou had achieved positive results. The high-value areas are concentrated in the northern part of the lower urban area, the northern part of the West Lake District, the northern part of the Binjiang District along the river, and the eastern part of the Jianggan District. Most of the communities in these areas are seriously aging communities with an elderly population of more than 1000 people. Communities with a high density of elderly groups mostly achieved good evaluation results, indicating that the construction, classification, and layout of grassroots public service facilities in different life circles are relatively good. In addition, high-value distribution not only occurs in the urban center, but also in the areas of Dangwangtou Community, Dieyuan Community, and Tangjiajing Community, which are relatively close to the edge of the city. Although the proportion of the elderly population in these communities is not high, the layout system of public service facilities is reasonable, and the construction concept is more advanced. Therefore, the evaluation results of suitability for the elderly were very good (Figure 20).

**Table 15.** Results of comprehensive elderly-adaptability evaluation of the community life circle.

| Community Life Circle Elderly-Adaptability Evaluation | Very High Values | High Values | Medium Value | Low Value | Very Low Value |
|---|---|---|---|---|---|
| Population cell (pcs) | 106 | 444 | 553 | 44 | 926 |
| Percentage of population cells (%) | 5.1 | 21.4 | 26.7 | 2.1 | 44.7 |

4.3.3. Comparative Analysis of Public Service Facilities Evaluation and Elderly Satisfaction

The statistical analysis of the elderly population for "Health Care for the Elderly, Providing for the Elderly, Achieving for the Elderly" facility use and "5-min, 10-min, and 15-min life circles" of the elderly construction satisfaction, compared with the above evaluation results, verified that the facility layout of the elderly evaluation system pattern is reasonable (Figure 14). The results showed that the equity of the facility layout of "Health Care for the Elderly, Providing for the Elderly, Achieving for the Elderly" and the evaluation results of "5-min, 10-min, and 15-min life circles" were positively correlated with the subjective satisfaction of the elderly(Figure 21). That is, the higher the level of the elderly, the higher their satisfaction. To a certain extent, this showed that from the perspective of health equity, it is reasonable to comprehensively analyze the layout of public facilities in the community life circle from both ends of supply and demand. However,

many factors still affect the satisfaction of the elderly. In addition to the equity of the layout of the facilities and the suitability of the life circle for the elderly, their health level, facility serviceability, social and economic factors, etc., affect satisfaction. This study can be used as an objective evaluation method to fill the gaps and find the possible shortcomings in the construction of healthy aging.

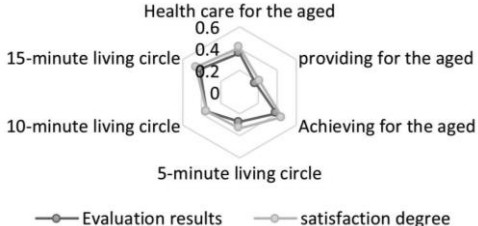

**Figure 21.** Comparison of elderly-adaptability evaluation results and satisfaction evaluation.

### 4.4. Identification of Four Spatial Differentiation Patterns

From the perspective of the interaction between supply and demand, four different matching patterns of supply and demand were formed by the spatial superposition of the demand of the elderly population and the comprehensive evaluation results of the supply of community life circle facilities for the elderly (Table 16):

- Balance between supply and demand (high demand of the elderly population and high supply of public service facilities);
- Lack of supply (high demand of the elderly population, low supply of public service facilities);
- Demand gap (low demand of the elderly population and high supply of public service facilities);
- Low supply and demand (low demand of the elderly population, low supply of public service facilities).

**Table 16.** Quantity statistics of different supply and demand matching patterns.

| Supply and Demand Matching Patterns | Population Cell (pcs) | Percentage of Population Cells (%) |
| --- | --- | --- |
| Balance between supply and demand (high demand-high supply) | 197 | 9.5 |
| Lack of supply (high demand-low supply) | 304 | 14.7 |
| Demand gap (low demand-high supply) | 352 | 17 |
| Low supply and demand (low demand-low supply) | 1219 | 58.8 |

Regarding the overall spatial distribution, the main urban area of Hangzhou showed a clear distribution pattern of high center and low edge. The high elderly population demand and high facility supply were mostly concentrated in the city's central area. The urban fringe area mainly had a low population demand and low facility supply (Figures 22 and 23).

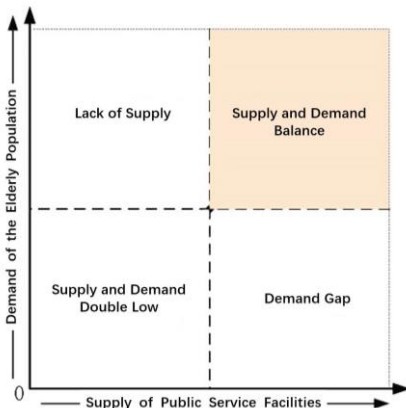

**Figure 22.** Diagram of supply and demand differential pattern.

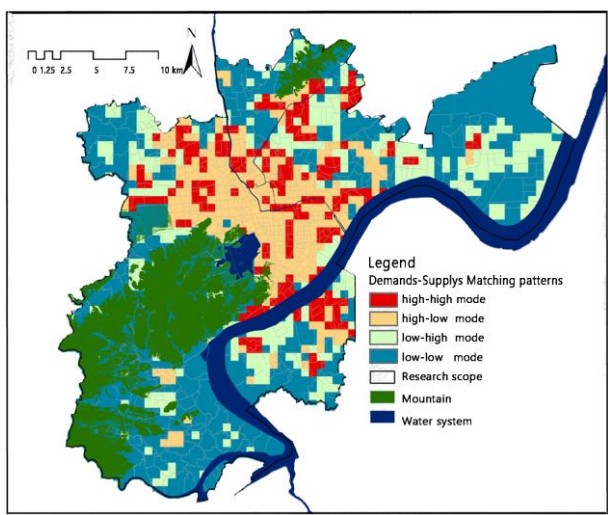

**Figure 23.** Spatial distribution of demand and supply matching patterns.

## 5. Discussion

### 5.1. Study on the Formation Mechanism of Spatial Differentiation

Through the above supply and demand balance (high-high pattern), lack of supply (high-low pattern), demand gap (low-high pattern), and supply and demand double low (low-low pattern), four supply and demand pattern matching space types were identified: resource replacement type, facilities lag type, development advanced type, and configuration blind area type.

The resource renewal replacement type corresponds to the central urban area with a dense elderly population and a good resource base. The follow-up development and construction mainly focus on the intensive updating of the stock. By integrating medical, pension, and sports facilities resources, combined with orderly community renewal, we could accurately supplement facilities with, for example, home care service points, outdoor fitness points, and small park green spaces. The facilities lag type corresponds to the core area of the old town. The elderly population base here is large, but the community area is small. The community is too old to update, and requires the replacement and upgrading of facilities to supplement and improve the number of them. New development corresponding to the construction of community and public service facilities configuration specifications is required. Subsequent replacement updates will be less complicated, and the sharing and diversification level will improve. The suburban configuration blind type corresponds to the suburban rural areas. In these areas, there are a large number of low-value blind spots suitable for the elderly. Health care, pension services, entertainment, and other public service facilities are in an extremely scarce state, although due to the small

number of elderly people, there is low demand. Nevertheless, in the long run, we must also fill gaps in pension demand facilities and improve the rural pension service system.

*5.2. Targeted Aging-Friendly Promotion Strategy*

In summarizing the characteristics and formation mechanism of the above spatial differentiation, from the perspective of the supply and demand interaction, it is necessary to put forward complementary strategies for these differentiated spaces to improve their internal health equity and aging level. The main optimization strategies are as follows:

- Continuous update optimization, focusing on facilities' quality improvement (for the supply and demand balance pattern).

It is necessary to monitor the needs of the elderly population in the region regularly and dynamically, and put planning strategies in place to prevent possible future supply and demand imbalances. The demand for care services for the elderly is increasing and the demand dimension is becoming more and more diversified, so the improvement of facilities suitable for the elderly can no longer be accomplished by simple measures, such as simply setting up new facilities for the elderly and adding barrier-free design. Development for the elderly should be characterized by sharing, comprehensiveness, and adaptability. By improving the quality of public service facilities, the elderly can avoid the "island effect", which refers to the inability to integrate into urban life and the feeling of disconnection from society in the process of using the facilities. We must continue to optimize and update facilities to improve the health equity of the elderly and the overall level of social aging.

- Deployment of facility configuration accuracy to ensure that the basic demands of the elderly are met (for the lack of supply pattern).

In view of the large number of elderly people in the core area of the old town and the high population density, it is necessary to start from the demand side, taking the actual health care needs of the elderly group as the starting point, and filling the supply gap of public service facilities from the bottom up through a more refined layout, so as to improve the construction of the pension infrastructure service system.

- Enrich the supply of facilities at all levels and improve the pension service system (for the demand gap pattern).

The more advanced layout pattern of public service facilities in the new town will be based on evaluation and feedback on the level of community life circle and pension demand. An abundant supply of public service facilities can better meet the health needs of the elderly residents in the corresponding area, and form a multi-level service system in medical care, pension, culture, entertainment, and health.

- Supplement the gap in facilities configuration and eliminate the blind area of pension services (for the low supply and demand pattern).

The existence of blind spots in pension services indicates that there is still a large number of elderly people in Hangzhou unable to meet their basic needs due to a low supply of facilities and low demand of the elderly population. There is a need to fill the blind spots of pension services to protect the basic medical, pension, and fitness needs of these people.

*5.3. Deficiencies and Prospects*

There are areas for improvement in this study. First, the lack of a comprehensive evaluation pattern, such as the quality of the facility itself and the level of service, is a limitation. Secondly, we did not explore health equity for other vulnerable groups in society, such as low-income people, people with disabilities, and children. Health equity advocates equal opportunities for all groups. Third, the study was subject to time limitations and limited questionnaire coverage. In addition, there are still some limitations and regional differences in strategy improvement. This paper took Hangzhou as an example, but failed to obtain more universal suggestions and strategies through a comparative analysis

between different cities to provide a more practical reference for improving the adaptive development of public service facilities for the elderly.

## 6. Conclusions

### 6.1. Evaluation Pattern of Public Service Facilities for the Elderly Based on the Perspective of Supply and Demand

In this paper, the two-step floating catchment area method was used to measure the service capacity of the facilities from both the supply and demand side in two steps, and the Gaussian attenuation function was added in the calculation process to more accurately simulate the attenuation caused by the actual service capacity of the facilities due to distance. Secondly, using the AHP analytic hierarchy process, the judgment matrix was established at two levels, and the public service facility health importance evaluation index table was calculated. Comprehensive analysis was carried out on the "Health Care for the Elderly, Providing for the Elderly, Achieving for the Elderly" facilities and the "5-min, 10-min, and 15-min life circles" of the elderly. Finally, the comprehensive evaluation results of the adaptability of the elderly in the community life circle were compared with the satisfaction evaluation of the elderly collected by a questionnaire survey, and we found that there was a significant positive correlation, which confirmed the accuracy and rationality of the evaluation pattern of the adaptability of the elderly.

### 6.2. Summary of the Problems in the Care Service System for the Elderly in the Main Urban Area of Hangzhou Using the Patterns of Public Service Facilities

Taking the main urban area of Hangzhou as a case study, and using the constructed evaluation patterns of public service facilities for the elderly, the problems in the current healthcare service system were determined.

- There is a significant gap in the old-age service facilities in the main urban area of Hangzhou. There is still much room for improvement. The lack of old-age service facilities and an old-age service system is the main problem.
- The 5-min and 10-min life circles must be improved for the elderly. For elderly people of advanced age who may have physical disabilities and limited walking ability, the layout of the 5-min and 10-min basic life circles' facilities is not reasonable, and the resource allocation system for facilities such as medical and health care, pension services, culture, and sports needs to be further improved.
- The allocation of public service facilities in the urban center and fringe areas needs to be balanced, as the current polarization seriously undermines health equity. There are blind spots in all kinds of facilities and life circles at all levels in the urban fringe area. The supply of facilities is insufficient, and the radiation range of pension services does not cover the elderly population in some areas, leading to poor adaptability and diversity.

### 6.3. Summary of the Spatial Differentiation Identification and Mechanism for the Elderly under the Interaction between Supply and Demand and the Resulting Optimization Strategy

Through the spatial superposition of the evaluation results of the actual demand and supply of the elderly, the four different spatial differentiation scenarios of supply and demand balance, supply shortage, demand gap, supply and demand double low were found. Aiming at the four spatial types, a refined promotion strategy can be proposed in terms of micro-layout. From a macro perspective, two points are proposed:

- From single planning indicators to a multi-dimensional demand for care services for the elderly

The optimization of the layout of public service facilities for the elderly needs to shift from purely supply-side planning quantitative indicators to multi-dimensional, people-oriented, and refined demand-side improvement, as the critical service population of public service facilities, the aged group, has many special needs compared with other age groups. The "Healthy China Action (2019–2030)" and "Healthy China 2030" plans put forward

multi-level requirements at the national level. Given the previous research analysis and development of "Elderly-Adaptability", it is clear that the single and isolated pension pattern should be adapted into an adaptive, diversified, and shared pension service system. It is necessary to transform the original single static thinking pattern into a dynamic and refined planning perspective.

- From a single to full demand coverage pattern

This article analyzed the types of public service facilities corresponding to the demands of the elderly, and the health characteristics that can be met. We need to improve the multi-dimensional needs in the matching process with the needs of the elderly. The facility layer configuration needs to change to meet social needs, self-realization, and other high-level needs, without public service facilities, through quality optimization under an intensive sharing mechanism.

Each community should utilize its advantages, actively carry out healthy cell engineering construction based on stock renewal, and create a healthy and supportive environment. Hangzhou has the highest density of elderly people, and it is challenging to update residential areas and public facilities. Therefore, it is necessary to undertake intensification and sharing as the primary development strategy to meet the needs of in situ pensions and community pensions as much as possible. The concept of intensive sharing must be implanted into the stock renewal system to better meet the diversified health care needs of the elderly through a more rational layout and combination of public service facilities.

**Author Contributions:** Conceptualization, Y.L. and Q.R.; methodology, Q.R.; formal analysis, L.D.; investigation, S.Y. and L.D.; resources, S.Y. and L.D.; data curation, L.D.; writing—original draft preparation, Q.R. and L.D.; writing—review and editing, Y.L., Q.R. and S.Y.; visualization, S.Y. and L.D.; supervision, Y.L.; funding acquisition, Y.L., S.Y. and L.D. All authors have read and agreed to the published version of the manuscript.

**Funding:** This research was funded by the National Natural Science Foundation of China (Grant No. 51878593), and the Center for Balance Architecture, Zhejiang University (Grant No. KH-20212946).

**Institutional Review Board Statement:** Not applicable.

**Informed Consent Statement:** Not applicable.

**Data Availability Statement:** The data presented in this study are available upon request from the authors.

**Acknowledgments:** The authors gratefully acknowledge the support of the funders.

**Conflicts of Interest:** The authors declare no conflict of interest.

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
