# Peer review of "Evaluation and Optimization of the Layout of Community Public Service Facilities for the Elderly: A Case Study of Hangzhou"

_land, doi:10.3390/land12030629_

Round 1

Reviewer 1 Report (Previous Reviewer 1)

The paper investigates the spatial distribution of senior community service centres in Hangzhou China and found that the senior facilities concentrate in the centre of the city where the demand is also high. A large amount of work has been done. There are some great pattern maps in the results section. However, the paper is structured poorly. It is recommended to correct the following issues before considering publishing.

1) the abstract needs to be shortened. Try to focus on the research question, method and key results. 

2) try to separate the introduction and literature review into two sections if possible. The introduction needs to give background on why is important is the research topic and what research questions the study tried to answer

3) the methodology section is a chaos. There is no figure 2.1 and I failed to understand the data in 2.2. It is suggested to straighten the section in conceptual framework, measures, study area and data aquisition

4) I did not see where section 5 starts. Is it possible to merge section 4 and 5 together?

5) move the discussion ahead of conclusion and focus on the finding of the study in the discussion.

Author Response

感谢您的建议!请参阅附件。

Reviewer 2 Report (Previous Reviewer 2)

Dear authors,

Thank you again for your re-submitted.

I checked the coverletter and the revised manuscript carefully.

For the latest version, I have a few comments.

If my comments would be considered properly, I would have no further comments for this manuscript.

Please see the followings: 

[1] All of the letters in your figure are too small to say, please re-layout or -redraw your figures, and make sure the information in your figures is clear.

[2] Chapter 6, the conclusion should be conclusions.

[3] Normally, the discussion chapter should be before the conclusions chapter.

[4] For the English editing, I suggest you find an English native speaker to edit, not yourself. There are many companies that have this kind of service. It would help you to improve the reading experience for this manuscript.

Round 2

Reviewer 1 Report (Previous Reviewer 1)

The authors made improvements to the manuscript based on the previous round review. It is recommended to publish after language polish and format check.

This manuscript is a resubmission of an earlier submission. The following is a list of the peer review reports and author responses from that submission.

Round 1

Reviewer 1 Report

The paper uses spatial analysis techniques to explore the layout of public facilities for seniors in Hangzhou, China. A significant amount of data were collected, compiled and analyzed. However, the paper has some mild structuring and formatting issues that need to be corrected before considering for publication.

1) the paper needs to separate the introduction and literature review. It is hard to understand the study question and goals. It is recommended to merge the literature review into the methodology section

2) The authors need to clarify the sources of some of the definitions, such as where figure 1.1 come from, who defined "health fairness" and so on

3) also, the method section broke down the facilities into different groups with some quotes. where are the quotes come from? the existing policy document, or interviews?

4)The results contain a large number of figures and tables. it is hard to follow. It is suggested to demonstrate the key findings instead of showing them all

5) the authors need to add a conclusion section to summarize the findings of the study

6) there are significant amount formatting issues that need to correct, such as broken sentences in line 120, lines 242-258 and so on. The maps need to clean, please get rid of the Chinese characters and the grey background. Also, clarify the "boundary" means. Figure 3.3 to 3.9 need to clearly label the exact map, such as which one is 5 minutes, 20 minutes and so on.

Author Response

Thank you very much for your valuable comments! Our reply is in the attachment.

Reviewer 2 Report

Dear authors,

Firstly, I would like to thank you for your work in this field.

After extensive reading, I only have a few comments for the current version, which are not many but essential.

For revision details, please see the followings:

[1] There are oceans of format problems in this paper. Please check it carefully and strictly follow the journal template. i.e. there are two "::" in your title. this kind of problem gives reviewers a bad reading experience. Please thoroughly check your paper, one word by one word;

[2] This paper lacks essential references. Normally, for the length of this paper, there should be at least 50 references. but now, the current number is 18. please double-check your statement and citation;

[3] I suggest you add more keywords. It will help this paper gets more exposure;

[4] Where is the source of figure 1.1 in the original text? You forgot to leave a citation;

[5] The English Language Editing for this paper is mandatory. the current version seems like translated by an online platform directly;

[6] What's the main innovation of this paper? Please point it out directly in the abstract;

[7] The figure 2.1 is good, but I can not find the relationship between it with your research. Please clarify it further. Also, please check the format of all of your figures and tables. and provide a clear figure with higher resolution;

[8] The research data source of this research, please point out what I need to follow and check your data. online document in English, thanks;

[9] You presented many data, but I do not see the logic between your chapter 3 and chapter 4. How did you get the context of the chapter 4 through chapter 3? By mathematical calculation? by statistical analysis? or by value evaluation? no analysis process explains;

[10] Normally, there should be a chapter named "conclusions".

Again, I have a high comment on your contribution.

The upon-revision recommendations are helping this manuscript be more readable for readers.

Author Response

Thank you very much for your valuable suggestions! Please see the attachment.

Reviewer 3 Report

Dear author/s,

the topic of the manuscript is interesting, however there are several aspects that must be improved:

1. The abstract is too general. The abstract should present the aim, methodology, main results, conclusions and implications of the study.

2. please state the research questions/hypotheses based on the literature review section. At the same time I recommend to emphasize what gap in the literature this study fills.

3. The data presented in table 3.2 are not clear. The category "Elderly people aged 60 and above" include also "Older persons 65 years and over" and so on?  If not please reconsider the superior limit of each group. It yea, please pay attention to the total number of the population.

4. What method was used to analyzed the data? Please clearly present it.

5. The discussion section must be improved. The results should be compared with other similar studies.

Others: pay attention to the format

Good luck!

Author Response

Thank you for your insightful comments!
